# FACTORED-NEUS: RECONSTRUCTING SURFACES, ILLUMINATION, AND MATERIALS OF POSSIBLY GLOSSY OBJECTS

## ABSTRACT

We develop a method that recovers the surface, materials, and illumination of a scene from its posed multi-view images. In contrast to prior work, it does not require any additional data and can handle glossy objects or bright lighting. It is a progressive inverse rendering approach, which consists of three stages. In the first stage, we reconstruct the scene radiance and signed distance function (SDF) with a novel regularization strategy for specular reflections. Our approach considers both volume and surface rendering, which allows for handling complex view-dependent lighting effects for surface reconstruction. In the second stage, we distill light visibility and indirect illumination from the learned SDF and radiance field using learnable mapping functions. Finally, we design a method for estimating the ratio of incoming direct light reflected in a specular manner and use it to reconstruct the materials and direct illumination. Experimental results demonstrate that the proposed method outperforms the current state-of-the-art in recovering surfaces, materials, and lighting without relying on any additional data.

## 1 INTRODUCTION

Reconstructing shape, material, and lighting from multiple views has wide applications in computer vision, virtual reality, augmented reality, and shape analysis. The emergence of neural radiance fields (Mildenhall et al., 2020) provides a framework for high-quality scene reconstruction. Subsequently, many works (Oechsle et al., 2021; Wang et al., 2021; Yariv et al., 2021; Wang et al., 2022; Fu et al., 2022) have incorporated implicit neural surfaces into neural radiance fields, further enhancing the quality of surface reconstruction from multi-views. Recently, several works (Munkberg et al., 2022; Zhang et al., 2021b;a; 2022b) have utilized coordinate-based networks to predict materials and learned parameters to represent illumination, followed by synthesizing image color using physically-based rendering equations to achieve material and lighting reconstruction. However, these methods typically do not fully consider the interdependence between different components, leading to the following issues with glossy surfaces when using real data.

First, surfaces with glossy materials typically result in highlights. The best current methods for reconstructing implicit neural surfaces rarely consider material information and directly reconstruct surfaces. The surface parameters can then be frozen for subsequent material reconstruction. Since neural radiance fields typically model such inconsistent colors as bumpy surfaces as shown in Fig. 1 left, the artifacts from surface reconstruction will affect material reconstruction if surfaces and materials are reconstructed sequentially. Second, a glossy surface can affect the decomposition of the reflected radiance into a diffuse component and a specular component. Typically, the specular component leaks into the diffuse component, resulting in inaccurate modeling as shown in Fig. 1 right. Third, focusing on synthetic data makes it easier to incorporate complex physically-based rendering algorithms, but they may not be robust enough to work on real data.

In this work, we consider the impact of glossy surfaces on surface and material reconstruction. To better handle glossy surfaces, we jointly use surface and volume rendering. Volume rendering does not decompose the reflected radiance, while surface rendering considers the diffuse and specular radiance separately. This approach better regularizes not only the decomposition of reflected light but also the surface reconstruction. In order to better recover diffuse and specular components, we estimate

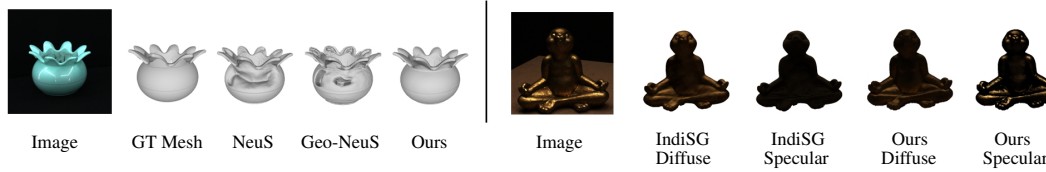

| Image | GT Mesh | NeuS | Geo-NeuS | Ours | Image | IndiSG Diffuse | IndiSG Specular | Ours Diffuse | Ours Specular |

Figure 1: Left: Geometry visualization for NeuS, Geo-NeuS and our method on the Pot scene from SK3D. Existing surface reconstruction methods struggle to recover the correct geometry of glossy objects due to the complex view-dependent effects they induce. The weak color model of these methods compels to represent such effects through concave geometric deformations rather than proper view-dependent radiance, leading to shape artifacts. In contrast, our method is able to correctly reconstruct a highly reflective surface due to our superior diffuse and specular color decomposition strategy. Right: Visualization of the recovered diffuse color component on the Bunny scene from DTU for IndiSG (Zhang et al., 2022b) and our method. Existing inverse rendering methods overestimate the diffuse material component in the presence of specular highlights. Our regularization strategy allows us to properly disentangle the color into diffuse and specular components.

the ratio of incoming light reflected in a specular manner. By introducing this parameter into a Spherical Gaussian representation of the BRDF, we can better model the reflection of glossy surfaces and decompose more accurate diffuse albedo information. Furthermore, we propose predicting continuous light visibility for signed distance functions to further enhance the quality of reconstructed materials and illumination. Our experimental results have shown that our factorization of surface, materials, and illumination achieves state-of-the-art performance on both synthetic and real datasets. Our main contribution is that we improve surface, material, and lighting reconstruction compared to PhySG (Zhang et al., 2021a), NVDiffRec (Munkberg et al., 2022), and IndiSG (Zhang et al., 2022b), the leading published competitors.

We believe that the good results of our approach compared to much recently published and unpublished work in material reconstruction is that we primarily developed our method on real data. The fundamental challenge for working on material and lighting reconstruction is the lack of available ground truth information for real datasets. Our solution to this problem was to work with real data and try to improve surface reconstruction as our main metric by experimenting with different materials and lighting decompositions as a regularizer. While we could not really measure the success of the material and lighting reconstruction directly, we could indirectly observe improvements in the surface metrics. By contrast, most recent and concurrent work uses surface reconstruction and real data more as an afterthought. This alternative route is to first focus on developing increasingly complex material and lighting reconstruction on synthetic data. However, we believe that this typically does not translate as well to real data as our approach.

## 2 RELATED WORK

**Neural radiance fields**. NeRF (Mildenhall et al., 2020) is a seminal work in 3D reconstruction. Important improvements were proposed by Mip-NeRF (Barron et al., 2021) and Mip-NeRF360 (Barron et al., 2022). One line of work explores the combination of different data structures with MLPs, such as factored volumes (Chan et al., 2022; Chen et al., 2022; Wang et al., 2023) or voxels (Müller et al., 2022; Reiser et al., 2021; Yu et al., 2021). There are multiple approaches that take a step towards extending neural radiance fields to reconstruct material information (Guo et al., 2022; Verbin et al., 2022; Ge et al., 2023; Yariv et al., 2023).

**Implicit neural surfaces**. Implicit neural surfaces are typically represented by occupancy functions or signed distance fields (SDFs). Some early works (Chen & Zhang, 2019; Mescheder et al., 2019; Park et al., 2019) take point clouds as input and output implicit neural surface representations. Many works have studied how to obtain implicit neural surfaces from images, initially focusing on surface rendering only (Niemeyer et al., 2020; Yariv et al., 2020). Subsequent methods followed NeRF to employ volume rendering, e.g. UNISURF (Oechsle et al., 2021), VolSDF (Yariv et al., 2021), NeuS (Wang et al., 2021), HF-NeuS (Wang et al., 2022), and Geo-NeuS (Fu et al., 2022).

**Joint reconstruction of surface, material, and illumination**. Ideally, we would like to jointly reconstruct the 3D geometry, material properties, and lighting conditions of a scene from 2D images. Several methods employ strategies to simplify the problem such as assuming known lighting condi-

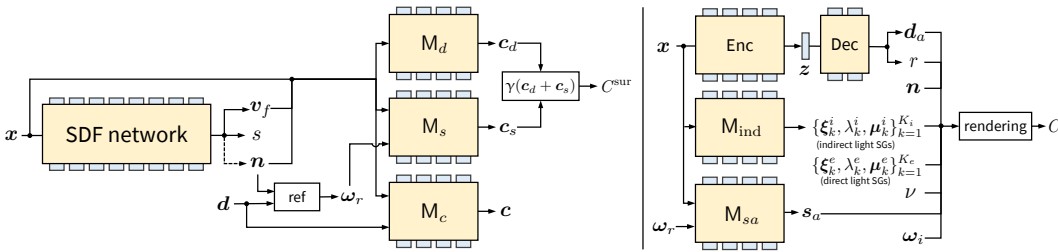

Figure 2: Overview for Stage 1 (left) and Stage 3 (right) training pipelines (Stage 2 pipeline is omitted due to its simplicity — see Sec 3.2 for details). The first stage (left) trains the SDF network $\mathsf{S}_\theta$ which outputs a feature vector $\boldsymbol{v}_f \in \mathbb{R}^{256}$, SDF value $s \in \mathbb{R}$, and normal $\boldsymbol{n} \in \mathbb{S}^2$ (as a normalized gradient of $s$; denoted via the dashed line); diffuse and specular surface color networks $\mathsf{M}_d$ and $\mathsf{M}_s$ produce their respective colors $\boldsymbol{c}_d, \boldsymbol{c}_s \in \mathbb{R}^3$ via surface rendering, which are then combined through tone mapping $\gamma(\cdot)$ to get the final surface color $C^{\mathrm{sur}} \in \mathbb{R}^3$; volumetric color network $\mathsf{M}_c$ produces the volumetrically rendered color $C^{\mathrm{vol}} \in \mathbb{R}^3$. The ref operation denotes computation of the reflection direction $\boldsymbol{\omega}_r \in \mathbb{S}^2$ from normal $\boldsymbol{n}$ and ray direction $\boldsymbol{\omega} \in \mathbb{S}^2$. In the third stage (right), we optimize the material BRDF auto-encoder with the sparsity constraint (Zhang et al., 2022b), our novel specular albedo network $\mathsf{M}_{sa}$, and the indirect illumination network $\mathsf{M}_{\mathrm{ind}}$. See Sec 3 for details.

tions (NeRV (Srinivasan et al., 2021) and NeRD (Boss et al., 2021a)) or pre-training (ENVIDR (Liang et al., 2023)). PhySG (Zhang et al., 2021a), NeRFactor (Zhang et al., 2021b), and NeROIC (Kuang et al., 2022), DIP (Deng et al., 2022) use Spherical Gaussians, point light sources, and spherical harmonics, respectively, to decompose unknown lighting from a set of images. Using an illumination integration network, Neural-PIL (Boss et al., 2021b) further reduces the computational cost of lighting integration. IRON (Zhang et al., 2022a) uses SDF-based volume rendering methods to obtain better geometric details in the shape recovery stage. NVDiffrec (Munkberg et al., 2022) explicitly extracts triangle mesh from tetrahedral representation for better material and lighting modeling. IndiSG (Zhang et al., 2022b) uses Spherical Gaussians to represent indirect illumination and achieves good lighting decomposition results. Some concurrent works (Jin et al., 2023; Wu et al., 2023a; Zhang et al., 2023a;b) continue to improve the efficiency and quality of inverse rendering but do not consider cases with a glossy appearance. NeAI (Zhuang et al., 2023) proposes neural ambient illumination to enhance the rendering quality of glossy appearance. Despite a lot of recent activity in this area, existing frameworks still struggle to effectively reconstruct reflective or glossy surfaces, lighting, and material information directly from images, especially real-world captured images. Appx Tab. 4 provides a comprehensive overview of recent inverse rendering techniques.

## 3 METHOD

Our framework has three training stages to gradually decompose the shape, materials, and illumination. The input to our framework is a set of images. In the first stage, we reconstruct the surface from a (possibly glossy) appearance decomposing the color into diffuse and specular components. After that, we use the reconstructed radiance field to extract direct illumination visibility and indirect illumination in the second stage. Having them decomposed from the radiance field allows for the recovery of the direct illumination map and materials' bidirectional reflectance distribution function (BRDF), which we perform in the final stage.

### 3.1 STAGE 1: SURFACE RECONSTRUCTION FROM GLOSSY APPEARANCE

Current inverse rendering methods first recover implicit neural surfaces, typically represented as SDFs, from multi-view images to recover shape information, then freeze the parameters of neural surfaces to further recover the material. However, this approach does not consider specular reflections that produce highlights and often models this inconsistent color as bumpy surface geometry as depicted in Fig. 1. This incorrect surface reconstruction has a negative impact on subsequent material reconstruction. We propose a neural surface reconstruction method that considers the appearance, diffuse color, and specular color of glossy surfaces at the same time, whose architecture is given in Fig. 2. Our inspiration comes from the following observations. First, according to Geo-NeuS, using SDF point cloud supervision can make the colors of surface points and volume rendering more

similar. We abandoned the idea of using additional surface points to supervise SDFs and directly use two different MLPs to predict the surface rendering and volume rendering results, and narrow the gap between these two colors using network training. In addition, when modeling glossy surfaces, Ref-NeRF proposes a method of decomposing appearance into diffuse and specular components, which can better model the glossy appearance. We propose to simultaneously optimize the radiance from the volumetric rendering and the surface rendering. For surface rendering, we further split the reflected radiance into a diffuse and a specular component. This can achieve an improved surface reconstruction of glossy surfaces.

**Shape representation**. We model shape as a signed distance function $\mathsf{S}_\theta : \boldsymbol{x} \mapsto (s, \boldsymbol{v}_f)$, which maps a 3D point $\boldsymbol{x} \in \mathbb{R}^3$ to its signed distance value $s \in \mathbb{R}$ and a feature vector $\boldsymbol{v}_f \in \mathbb{R}^{256}$. SDF allows computing a normal $\boldsymbol{n}$ directly by calculating the gradient: $\boldsymbol{n} = \nabla \mathsf{S}_\theta(\boldsymbol{x}) / \|\nabla \mathsf{S}_\theta(\boldsymbol{x})\|$.

**Synthesize appearance.** Learning implicit neural surfaces from multi-view images often requires synthesizing appearance to optimize the underlying surface. The recent use of volume rendering in NeuS (Wang et al., 2021) has been shown to better reconstruct surfaces. According to Eq. 14 in Appx A, the discretization formula for volume rendering is $C^{\text{vol}} = \sum_{i=1}^{n} T_i \alpha_i \boldsymbol{c_i} = \sum_{i=1}^{n} w_i \boldsymbol{c_i}$ with $n$ sampled points $\{\boldsymbol{r}(t_i)\}_{i=1}^{n}$ on the ray. where $\alpha_i = \max(\Phi_s(\mathsf{S}_\theta(\boldsymbol{r}(t_i))) - \Phi_s(\mathsf{S}_\theta(\boldsymbol{r}(t_{i+1}))) / \Phi_s(\mathsf{S}_\theta(\boldsymbol{r}(t_i))), 0)$, which is discrete opacity values following NeuS, where $\Phi_s = 1/(1 + e^{-x})$ is a sigmoid function and $T_i = \prod_{j=1}^{i-1}(1 - \alpha_j)$ is the discrete transparency. Similar to the continuous case, we can also define discrete weights $w_i = T_i \alpha_i$.

To compute color $\boldsymbol{c_i}$ on the point $\boldsymbol{r}(t_i)$, we define a color mapping $\mathsf{M}_c : (\boldsymbol{x}, \boldsymbol{n}, \boldsymbol{d}, \boldsymbol{v}_f) \mapsto \boldsymbol{c}$ from any 3D point $\boldsymbol{x}$ given its feature vector $\boldsymbol{v}_f$, normal $\boldsymbol{n}$ and ray direction $\boldsymbol{d}$.

**Synthesize diffuse and specular components.** In addition to synthesizing appearance, we also synthesize diffuse and specular components. This idea comes from surface rendering, which better handles surface reflections. From Eq. 16 in Appx A, the radiance $L_o$ of surface point $\boldsymbol{x}$ and outgoing viewing direction $\boldsymbol{\omega}_o$ can be decomposed into two parts: diffuse and specular radiance.

$$L_o(\boldsymbol{x}, \boldsymbol{\omega}_o) = \frac{\boldsymbol{d}_a}{\pi} \int_\Omega L_i(\boldsymbol{x}, \boldsymbol{\omega}_i)(\boldsymbol{\omega}_i \cdot \boldsymbol{n}) d\boldsymbol{\omega}_i \tag{1}$$

$$+ \int_\Omega f_s(\boldsymbol{x}, \boldsymbol{\omega}_i, \boldsymbol{\omega}_o) L_i(\boldsymbol{x}, \boldsymbol{\omega}_i)(\boldsymbol{\omega}_i \cdot \boldsymbol{n}) d\boldsymbol{\omega}_i \tag{2}$$

$$= \mathsf{M}_d(\boldsymbol{x}, \boldsymbol{n}) + \mathsf{M}_s(\boldsymbol{x}, \boldsymbol{\omega}_o, \boldsymbol{n}) \tag{3}$$

We define two neural networks to predict diffuse and specular components separately. We use the term diffuse radiance to refer to the component of the reflected radiance that stems from a diffuse surface reflection. We define a mapping $\mathsf{M}_d : (\boldsymbol{x}, \boldsymbol{n}, \boldsymbol{v}_f) \mapsto \boldsymbol{c}_d$ for diffuse radiance that maps surface points $\boldsymbol{x}$, surface normals $\boldsymbol{n}$, and feature vectors $\boldsymbol{v}_f$ to diffuse radiance. For simplicity, we assume that the diffuse radiance is not related to the outgoing viewing direction $\boldsymbol{\omega}_o$.

We use the term specular radiance to describe the non-diffuse (view-direction dependent) component of the reflected radiance. Ref-NeRF (Verbin et al., 2022) proposes to model the glossy appearance using the reflection direction instead of the viewing one. However, from Eq. 3, we can observe that specular radiance is also highly dependent on the surface normal, which is particularly important when reconstructing SDF. In contrast to Ref-NeRF, we further condition specular radiance on the surface normal. Therefore, we define specular radiance $\mathsf{M}_s : (\boldsymbol{x}, \boldsymbol{\omega}_r, \boldsymbol{n}, \boldsymbol{v}_f) \mapsto \boldsymbol{c}_s$, which maps surface points $\boldsymbol{x}$, outgoing viewing direction $\boldsymbol{\omega}_o$, surface normals $\boldsymbol{n}$, and feature vectors $\boldsymbol{v}_f$ to specular radiance, where $\boldsymbol{\omega}_r = 2(\boldsymbol{\omega}_o \cdot \boldsymbol{n})\boldsymbol{n} - \boldsymbol{\omega}_o$.

Surface rendering focuses the rendering process on the surface, allowing for a better understanding of highlights on the surface compared to volume rendering, but requires calculating surface points. We sample $n$ points on the ray $\{\boldsymbol{r}(t_i)|i = 1, ..., n\}$. We query the sampled points to find the first point $\boldsymbol{r}(t'_i)$ whose SDF value is less than zero $\mathsf{S}_\theta(\boldsymbol{r}(t'_i)) < 0$. Then the point $\boldsymbol{r}(t_{i'-1})$ sampled before $\boldsymbol{r}(t'_i)$ has the SDF value greater than or equal to zero $\mathsf{S}_\theta(\boldsymbol{r}(t_{i'-1})) \geq 0$. To account for the possibility of rays interacting with objects and having multiple intersection points, we select the first point with a negative SDF value to solve this issue.

We use two neural networks to predict the diffuse radiance and specular radiance of two sampling points $\boldsymbol{r}(t_{i'-1})$ and $\boldsymbol{r}(t'_i)$. The diffuse radiance of the two points calculated by the diffuse network $\mathsf{M}_d$ will be $\boldsymbol{c}_d^{i'-1}$ and $\boldsymbol{c}_d^{i'}$. The specular radiance of the two points calculated by the specular network

$M_s$ will be $c_s^{i'-1}$ and $c_s^{i'}$. Therefore, the diffuse radiance and specular radiance of the surface point $x$ can be calculated as follows.

$$c_d = M_d(x, n) = \frac{w_{i'-1}c_d^{i'-1} + w_i'c_d^{i'}}{w_{i'-1} + w_i'} \tag{4}$$

$$c_s = M_s(x, \omega_o, n) = \frac{w_{i'-1}c_s^{i'-1} + w_i'c_s^{i'}}{w_{i'-1} + w_i'} \tag{5}$$

The final radiance of the intersection of the ray and the surface is calculated by a tone mapping:

$$C^{\mathrm{sur}} = \gamma(c_d + c_s) \tag{6}$$

where $\gamma$ is a pre-defined tone mapping function that converts linear color to sRGB (Verbin et al., 2022) while ensuring that the resulting color values are within the valid range of [0, 1].

**Training strategies.** In our training process, we define three loss functions, namely volume radiance loss $\mathcal{L}_{\mathrm{vol}}$, surface radiance loss $\mathcal{L}_{\mathrm{sur}}$, and regularization loss $\mathcal{L}_{\mathrm{reg}}$. The volume radiance loss $\mathcal{L}_{\mathrm{vol}}$ is measured by calculating the $\mathcal{L}_1$ distance between the ground truth colors $C^{\mathrm{gt}}$ and the volume radiances $C^{\mathrm{vol}}$ of a subset of rays $\mathcal{R}$. The surface radiance loss $\mathcal{L}_{\mathrm{sur}}$ is measured by calculating the $\mathcal{L}_1$ distance between the ground truth colors $C^{\mathrm{gt}}$ and the surface radiances $C^{\mathrm{sur}}$. $\mathcal{L}_{\mathrm{reg}}$ is an Eikonal loss term on the sampled points. We use weights $\lambda_{\mathrm{sur}}$ and $\lambda_{\mathrm{reg}}$ to balance the impact of these three losses. The overall training loss is as follows. See Appx C for details of training strategies.

$$\mathcal{L} = \mathcal{L}_{\mathrm{vol}} + \lambda_{\mathrm{sur}}\mathcal{L}_{\mathrm{sur}} + \lambda_{\mathrm{reg}}\mathcal{L}_{\mathrm{reg}} \tag{7}$$

### 3.2 STAGE 2: LEARNING DIRECT LIGHTING VISIBILITY AND INDIRECT ILLUMINATION

At this stage, we focus on predicting the lighting visibility and indirect illumination of a surface point $x$ under different incoming light direction $\omega_i$ using the SDF in the first stage.

Visibility is an important factor in shadow computation. It calculates the visibility of the current surface point $x$ in the direction of the incoming light $\omega_i$. Path tracing of the SDF is commonly used to obtain a binary visibility (0 or 1) as used in IndiSG (Zhang et al., 2022b), but this kind of visibility is not friendly to network learning. Inspired by NeRFactor (Zhang et al., 2021b), we propose to use an integral representation with the continuous weight function $w(t)$ (from 0 to 1) for the SDF to express light visibility. Specifically, we establish a neural network $M_\nu : (x, \omega_i) \mapsto \nu$, that maps the surface point $x$ and incoming light direction $\omega_i$ to visibility, and the ground truth value of light visibility is obtained by integrating the weights $w_i$ of the SDF of sampling points along the incoming light direction and can be expressed as $\nu^{gt} = 1 - \sum_{i=1}^{n} w_i$.

Indirect illumination refers to the light that is reflected or emitted from surfaces in a scene and then illuminates other surfaces, rather than directly coming from a light source, which contributes to the realism of rendered images. Following IndiSG (Zhang et al., 2022b), we parameterize indirect illumination $I(x, \omega_i)$ via $K_i = 24$ Spherical Gaussians (SGs). For more details, see Appx D.

### 3.3 STAGE 3: RECOVERING MATERIALS AND DIRECT ILLUMINATION

Reconstructing good materials and lighting from scenes with highlights is a challenging task. Following prior works Zhang et al. (2022b; 2021a), we use the Disney BRDF model (Burley & Studios, 2012) and represent BRDF $f_s(\omega_i \mid \xi_s, \lambda_s, \mu_s)$ via Spherical Gaussians (Zhang et al., 2021a). Direct (environment) illumination is represented using $K_e = 128$ SGs:

$$E(x, \omega_i) = \sum_{k=1}^{K_e} E_k(\omega_i \mid \xi_k^e, \lambda_k^e, \mu_k^e) \tag{8}$$

and render diffuse radiance and specular radiance of direct illumination in a way similar to Eq. 2.

$$L_d(x) = \frac{d_a}{\pi} \sum_{k=1}^{K_e} (\nu(x, \omega_i) \otimes E_k(\omega_i)) \cdot (\omega_i \cdot n) \tag{9}$$

$$L_s(x, \omega_o) = \sum_{k=1}^{K_e} (f_s \otimes \nu(x, \omega_i) \otimes E_k(\omega_i)) \cdot (\omega_i \cdot n) \tag{10}$$

where $\boldsymbol{d}_a$ is diffuse albedo.

To reconstruct a more accurate specular reflection effect, we use an additional neural network $\mathsf{M}_{sa} : (\boldsymbol{x}, \boldsymbol{\omega}_r) \mapsto \boldsymbol{s}_a \in [0, 1]$ to predict the specular albedo. The modified BRDF $f_s^a$ is as follows:

$$f_s^a = \boldsymbol{s}_a \otimes f_s(\boldsymbol{\omega}_i; \boldsymbol{\xi}, \lambda, \boldsymbol{\mu}) = f_s(\boldsymbol{\omega}_i; \boldsymbol{\xi}, \lambda, \boldsymbol{s}_a\boldsymbol{\mu}) \tag{11}$$

For indirect illumination, the radiance is extracted directly from another surface and does not consider light visibility. The diffuse radiance and specular radiance of indirect illumination are as follows

$$L_d^{\text{ind}}(\boldsymbol{x}) = \frac{\boldsymbol{d}_a}{\pi} \sum_{k=1}^{T} I_k(\boldsymbol{x}, \boldsymbol{\omega}_i) \cdot (\boldsymbol{\omega}_i \cdot \boldsymbol{n}) \tag{12}$$

$$L_s^{\text{ind}}(\boldsymbol{x}, \boldsymbol{\omega}_o) = \sum_{k=1}^{T} (\boldsymbol{s}_a \otimes f_s) \otimes I_k(\boldsymbol{x}, \boldsymbol{\omega}_i) \cdot (\boldsymbol{\omega}_i \cdot \boldsymbol{n}) \tag{13}$$

Our final synthesized appearance is $C = L_d + L_s + L_d^{\text{ind}} + L_s^{\text{ind}}$ and supervised via an $\mathcal{L}_1$ RGB loss.

## 4 EXPERIMENTS

### 4.1 EVALUATION SETUP

**Datasets.** To evaluate the quality of surface reconstruction, we use the DTU (Jensen et al., 2014), SK3D (Voynov et al., 2022), and Shiny (Verbin et al., 2022) datasets. DTU and SK3D are two real-world captured datasets, while Shiny is synthetic. In DTU, each scene is captured by 49 or 64 views of 1600×1200 resolution. From this dataset, we select 3 scenes with specularities to verify our proposed method in terms of surface quality and material decomposition. In the SK3D dataset, the image resolution is 2368×1952, and 100 views are provided for each scene. This dataset contains more reflective objects with complex view-dependent lighting effects that pose difficulties in surface and material reconstruction. From SK3D, we select 4 glossy surface scenes with high levels of glare to validate our proposed method. The Shiny dataset has 5 different glossy objects rendered in Blender under conditions similar to NeRF's dataset (100 training and 200 testing images per scene). The resolution of each image is 800×800.

To evaluate the effectiveness of material and lighting reconstruction, we use the dataset provided by IndiSG (Zhang et al., 2022b), which has self-occlusion and complex materials. Each scene has 100 training images of $800 \times 800$ resolution. To evaluate the quality of material decomposition, the dataset also provides diffuse albedo, roughness, and masks for testing.

**Baselines.** Our main competitors are the methods that can also reconstruct all three scene properties: surface geometry, materials, and illumination. We choose NVDiffRec (Munkberg et al., 2022), PhySG (Zhang et al., 2021a), and IndiSG (Zhang et al., 2022b) due to their popularity and availability of the source code. NVDiffRec uses tetrahedral marching to extract triangle meshes and obtains good material decomposition using a triangle-based renderer. PhySG optimizes geometry and material information at the same time using a Spherical Gaussian representation for direct lighting and material. IndiSG first optimizes geometry and then uses a Spherical Gaussian representation for indirect lighting to improve the quality of material reconstruction.

Apart from that, we also compare against more specialized methods for individual quantitative and qualitative comparisons to provide additional context for our results. For surface reconstruction quality, we compare our method to NeuS (Wang et al., 2021) and Geo-NeuS (Fu et al., 2022). NeuS is a popular implicit surface reconstruction method that achieves strong results without reliance on extra data. Geo-NeuS improves upon NeuS by using additional point cloud supervision, obtained from structure from motion (SfM) (Schönberger & Frahm, 2016). We also show a qualitative comparison to Ref-NeRF (Verbin et al., 2022), which considers material decomposition, but due to modeling geometry using density function, it has difficulty extracting smooth geometry.

**Evaluation metrics.** We use the official evaluation protocol to compute the Chamfer distance (lower values are better) for the DTU dataset and also use the Chamfer distance for the SK3D dataset. We utilize the PSNR metric (higher values are better), to quantitatively evaluate the quality of rendering, material, and illumination. We follow IndiSG (Zhang et al., 2022b) and employ masks to compute the PSNR metric in the foreground to evaluate the quality of materials and rendering. See Appx B for implementation details.

Table 1: Quantitative results in terms of Chamfer distance on DTU (Jensen et al., 2014) and SK3D (Voynov et al., 2022).

| | DTU 63 | DTU 97 | DTU 110 | DTU 122 | **Mean** | Pot | Funnel | Snowman | Jug | **Mean** |
|---|---|---|---|---|---|---|---|---|---|---|
| NeuS Wang et al. (2021) | 1.01 | 1.21 | 1.14 | 0.54 | 0.98 | 2.09 | 3.93 | 1.40 | 1.81 | 2.31 |
| Geo-NeuS Fu et al. (2022) | **0.96** | **0.91** | **0.70** | **0.37** | **0.73** | 1.88 | 2.03 | 1.64 | 1.68 | 1.81 |
| PhySG Zhang et al. (2021a) | 4.16 | 4.99 | 3.57 | 1.42 | 3.53 | 14.40 | 7.39 | 1.55 | 7.59 | 7.73 |
| IndiSG Zhang et al. (2022b) | 1.15 | 2.07 | 2.60 | 0.61 | 1.61 | 5.62 | 4.05 | 1.74 | 2.35 | 3.44 |
| Factored-NeuS (ours) | 0.99 | 1.15 | 0.89 | 0.46 | 0.87 | **1.54** | **1.95** | **1.31** | **1.40** | **1.55** |

Figure 3: Qualitative results for DTU (left) and SK3D (right).

## 4.2 SURFACE RECONSTRUCTION QUALITY

We first demonstrate quantitative results in terms of Chamfer distance. IndiSG and PhySG share the same surface reconstruction method, but PhySG optimizes it together with the materials, while IndiSG freezes the underlying SDF after its initial optimization. We list the numerical results for IndiSG and PhySG for comparison. NVDiffrec is not as good for surface reconstruction as we verify qualitatively in Appx Fig. 7. For completeness, we also compare our method against NeuS and Geo-NeuS. First, we list quantitative results on the DTU dataset and SK3D dataset in Tab. 1. It should be noted that NeuS and Geo-NeuS can only reconstruct surfaces from multi-views, while our method and IndiSG can simultaneously tackle shape, material, and lighting. As shown in the table, Geo-NeuS achieves better performance on the DTU dataset because the additional sparse 3D points generated by structure from motion (SfM) for supervising the SDF network are accurate. Our approach can also incorporate the components of Geo-NeuS based on extra data, and the surface reconstruction quality will be further improved as shown in Appx F. However, on the SK3D scenes with glossy surfaces, these sparse 3D points cannot be generated accurately by SfM, leading to poor surface reconstruction by Geo-NeuS. In contrast, our approach can reconstruct glossy surfaces on both DTU and SK3D without any explicit geometry information. Compared with IndiSG, PhySG cannot optimize geometry and material information well simultaneously on real-world acquired datasets with complex lighting and materials. Our method is the overall best method on SK3D. Most importantly, we demonstrate large improvements over IndiSG and PhySG, our main competitors, on both DTU and SK3D. We further demonstrate the qualitative experimental comparison results in Fig. 3. It can be seen that although Geo-NeuS has the best quantitative evaluation metrics, it loses some of the fine details, such as the small dents on the metal can in DTU 97. By visualizing the results of the SK3D dataset, we can validate that our method can reconstruct glossy surfaces without explicit geometric supervision. See Appx Fig. 7 for the Shiny dataset.

## 4.3 MATERIAL RECONSTRUCTION AND RENDERING QUALITY

In Tab. 2, we evaluate the quantitative results in terms of PSNR metric for material and illumination reconstruction on the IndiSG dataset compared with PhySG, NVDiffrec, and IndiSG. For completeness, we also compare to the case where the specular albedo improvement was not used in Stage 3 (See in Eq. 11 in Section 3.3). Regarding diffuse albedo, although NVDiffrec showed a slight improvement over us in the balloons scene, we achieved a significant improvement over NVDiffrec in the other three scenes. Our method achieved the best results in material reconstruction. Moreover, our method achieves the best results in illumination quality without using the specular albedo improvement. Additionally, our method significantly outperforms other methods in terms of rendering quality and achieves better appearance synthesis results. We present the qualitative results of material reconstruction in Fig. 4, which shows that our method has better detail capture compared to IndiSG and PhySG, such as the text on the balloon. Although NVDiffrec can reconstruct the nails

Table 2: Quantitative results in terms of PSNR on IndiSG (Zhang et al., 2022b) dataset for IndiSG and our method. "SAI" refers to specular albedo improvement.

| | Baloons | | | Hotdog | | | Chair | | | Jugs | | | Mean | | |
| --- | --- | --- | --- | --- | --- | --- | --- | --- | --- | --- | --- | --- | --- | --- | --- |
| | albedo | illumination | rendering | albedo | illumination | rendering | albedo | illumination | rendering | albedo | illumination | rendering | albedo | illumination | rendering |
| PhySG | 15.91 | 13.89 | 27.83 | 13.95 | 11.69 | 25.13 | 14.86 | 12.26 | 28.32 | 16.84 | 10.92 | 28.20 | 15.39 | 12.19 | 27.37 |
| NVDiffrec | **26.88** | 14.63 | 29.90 | 13.60 | 22.43 | 33.68 | 21.12 | 15.56 | 29.16 | 11.20 | 10.47 | 25.30 | 20.41 | 13.56 | 29.51 |
| IndiSG | 21.95 | 25.24 | 24.40 | 26.43 | 21.87 | 31.77 | 24.71 | **22.17** | 24.98 | 21.44 | 20.59 | 24.91 | 23.63 | 22.47 | 26.51 |
| Ours *w/o* SAI | 24.09 | **25.97** | 28.82 | 30.58 | **23.50** | 36.05 | 25.23 | 22.13 | 32.64 | 19.64 | 20.40 | 33.56 | 24.89 | **23.00** | 32.77 |
| Ours | 25.79 | 21.79 | **33.89** | **30.72** | 20.23 | **36.71** | **26.33** | 20.97 | **34.58** | 22.94 | **21.84** | **36.48** | **26.28** | 21.21 | **35.41** |

Figure 4: Qualitative results on IndiSG dataset in terms of albedo reconstruction (left half) and novel view synthesis quality (right half).

on the backrest, its material decomposition effect is not realistic. The materials reconstructed by our method are closer to ground truth ones. We also demonstrate the material decomposition effectiveness of our method on Shiny datasets with glossy surfaces, as shown in Appx Fig. 9. We showcase the diffuse albedo and rendering results of NVDiffrec, IndiSG, and our method. The rendering results indicate that our method can restore the original appearance with specular highlights more accurately, such as the reflections on the helmet and toaster compared to the IndiSG and NVDiffrec methods. The material reconstruction results show that our diffuse albedo contains less specular reflection information compared to other methods, indicating our method has better ability to suppress decomposition ambiguity caused by specular highlights. We also provide qualitative results on real-world captured datasets such as DTU and SK3D in Appx Fig. 8. To offer a more detailed presentation of the reconstruction quality across continuous viewpoints, we include videos of diffuse albedo, indirect illumination, light visibility, and rendering for three different scenes in the supplementary materials. Furthermore, we perform relighting for these three scenes and provide videos to assess the relighting quality.

## 4.4 ABLATION STUDY

**Materials and illumination.** We conduct an ablation study on the different components we proposed by evaluating their material and lighting performance on a complex scene, the hotdog, as shown in Tab. 3. "SI" refers to surface improvement, which means using networks to synthesize diffuse and specular color at the same time. "VI" stands for visibility improvement, which involves continuous light visibility supervision based on the SDF. "SAI" refers to specular albedo improvement, which incorporates specular albedo into the BRDF of Spherical Gaussians. We compare different settings in terms of diffuse albedo, roughness, appearance synthesis, and illumination. We used IndiSG as a reference and find that introducing volume rendering can improve the accuracy of material and lighting reconstruction. When the surface has no defects, further performing the surface improvement will enhance the quality of roughness and rendering but may cause a decrease in lighting reconstruction quality. Making the visibility supervision continuous improves the reconstruction of diffuse albedo, roughness, and lighting, but it also affects rendering quality. Introducing specular albedo can greatly improve roughness and rendering quality but negatively affect lighting reconstruction quality. We further show qualitative results in Appx Fig. 10. It can be observed that after improving the light visibility, the white artifacts at the edges of the plate in diffuse albedo are significantly reduced. Introducing specular albedo also makes the sausage appear smoother

and closer to its true color roughness, represented by black. In terms of lighting, when not using specular albedo, the lighting reconstruction achieves the best result, indicating a clearer reconstruction of ambient illumination. In summary, our ablation study highlights the importance of taking into account various factors when reconstructing materials and illumination from images. By evaluating the performance of different modules, we can better understand their role in improving the reconstruction quality.

**Surface reconstruction.** To validate our surface reconstruction strategy in Stage 1, we selected the Pot scene from SK3D and ablated the method the following way. "1Vol + 1Sur" means that we only use volume rendering and surface rendering MLPs for surface reconstruction, without decomposing material information into diffuse and specular components. "1Vol + 2Vol" means we use two volume reconstructions where one of them is split into diffuse and specular components. Just using "2Vol" to split diffuse and specular components will fail to reconstruct the correct surface due to inaccurate normal vectors in reflection direction computation, especially when modeling objects with complex materials or lighting effects. We provide the quantitative (Chamfer distance) and qualitative results of different frameworks in Fig. 5. It can be seen that synchronizing the volume color and the color on the surface point has a certain effect in suppressing concavities, but still cannot meet the requirements for complex glossy surfaces with strong reflections. Using volume rendering to decompose diffuse and specular components can result in excessive influence from non-surface points, which still causes small concavities. When combining these two methods, our approach can achieve reconstruction results without concavities.

Table 3: Ablation study for materials and illumination decomposition in terms of PSNR. "Alb" stands for "diffuse albedo", "Rough" is "roughness", "Rend" is "appearance", and "Illu" is "illumination".

| Method | Alb | Rough | Rend | Illu |
|---|---|---|---|---|
| IndiSG Zhang et al. (2022b) | 26.44 | 15.97 | 31.78 | 21.88 |
| Ours *w/o* SAI *w/o* VI *w/o* SI | 29.31 | 16.98 | 35.48 | 23.48 |
| Ours *w/o* SAI *w/o* VI | 29.64 | 17.86 | 36.36 | 23.41 |
| Ours *w/o* SAI | 30.58 | 18.83 | 36.05 | **23.50** |
| Ours | **30.76** | **23.10** | **36.71** | 20.24 |

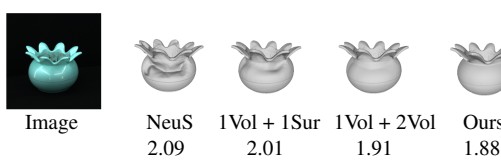

| Image | NeuS 2.09 | 1Vol + 1Sur 2.01 | 1Vol + 2Vol 1.91 | Ours 1.88 |

Figure 5: Ablation study for different surface reconstruction methods. See the previous table caption for an explanation.

## 5 CONCLUSIONS

In this work, we propose Factored-NeuS, a novel approach to inverse rendering that reconstructs geometry, material, and lighting from multiple views. Our first contribution is to simultaneously synthesize the appearance, diffuse radiance, and specular radiance during surface reconstruction, which allows the geometry to be unaffected by glossy highlights. Our second contribution is to train networks to estimate reflectance albedo and learn a visibility function supervised by continuous values based on the SDF, so that our method is capable of better decomposing material and lighting. Experimental results show that our method surpasses the state-of-the-art in both geometry reconstruction quality and material reconstruction quality. A future research direction is how to effectively decompose materials for fine structures, such as nails on the backrest of a chair.

In certain scenarios, our method still faces difficulties. For mesh reconstruction, we can only enhance results on scenes with smooth surfaces and few geometric features. Despite improvements on the glossy parts in the DTU 97 results, the overall Chamfer distance does not significantly decrease. As seen in Fig. 4, the reconstructed albedo of the chair still lacks some detail. The nails on the chair and the textures on the pillow are not accurately captured in the reconstructed geometry. Moreover, we do not foresee any negative societal implications directly linked to our research on surface reconstruction.

In future work, we would like to focus on the reconstruction of dynamic objects and humans. We also would like to include additional data acquisition modalities for improved performance.

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

## A    PRELIMINARIES

**Volume rendering**. The radiance $C$ of the pixel corresponding to a given ray $\boldsymbol{r}(t) = \boldsymbol{o} + t\boldsymbol{d}$ at the origin $\boldsymbol{o} \in \mathbb{R}^3$ towards direction $\boldsymbol{d} \in \mathbb{S}^2$ is calculated using the volume rendering equation, which involves an integral along the ray with boundaries $t_n$ and $t_f$ ($t_n$ and $t_f$ are parameters to define the near and far clipping plane). This calculation requires the knowledge of the volume density $\sigma$ and directional color $\boldsymbol{c}$ for each point within the volume.

$$C(\boldsymbol{r}) = \int_{t_n}^{t_f} T(t)\sigma(\boldsymbol{r}(t))\mathbf{c}(\boldsymbol{r}(t), \boldsymbol{d})dt \tag{14}$$

The volume density $\sigma$ is used to calculate the accumulated transmittance $T(t)$:

$$T(t) = \exp\left(-\int_{t_n}^{t_f} \sigma(\boldsymbol{r}_s)ds\right) \tag{15}$$

It is then used to compute a weighting function $w(t) = T(t)\sigma(\boldsymbol{r}(t))$ to weigh the sampled colors along the ray $\boldsymbol{r}(t)$ to integrate into radiance $C(\boldsymbol{r})$.

**Surface rendering**. The radiance $L_o(\boldsymbol{x}, \boldsymbol{\omega}_o)$ reflected from a surface point $\boldsymbol{x}$ in direction $\boldsymbol{\omega}_o = -\boldsymbol{d}$ is an integral of bidirectional reflectance distribution function (BRDF) and illumination over half sphere $\Omega$, centered at normal $\boldsymbol{n}$ of the surface point $\boldsymbol{x}$:

$$L_o(\boldsymbol{x}, \boldsymbol{\omega}_o) = \int_{\Omega} L_i(\boldsymbol{x}, \boldsymbol{\omega}_i) f_r(\boldsymbol{x}, \boldsymbol{\omega}_i, \boldsymbol{\omega}_o)(\boldsymbol{\omega}_i \cdot \boldsymbol{n})\mathrm{d}\boldsymbol{\omega}_i \tag{16}$$

where $L_i(\boldsymbol{x}, \boldsymbol{\omega}_i)$ is the illumination on $\boldsymbol{x}$ from the incoming light direction $\boldsymbol{\omega}_i$, and $f_r$ is BRDF, which is the proportion of light reflected from direction $\boldsymbol{\omega}_i$ towards direction $\boldsymbol{\omega}_o$ at the point $\boldsymbol{x}$.

## B    IMPLEMENTATION DETAILS

Our full model is composed of several MLP networks, each one of them having a width of 256 hidden units unless otherwise stated. In Stage 1, the SDF network $\mathsf{S}_\theta$ is composed of 8 layers and includes a skip connection at the 4-th layer, similar to NeuS (Wang et al., 2021). The input 3D coordinate $\boldsymbol{x}$ is encoded using positional encoding with 6 frequency scales. The diffuse color network $\mathsf{M}_d$ utilizes a 4-layer MLP, while the input surface normal $\boldsymbol{n}$ is positional-encoded using 4 scales. For the specular color network $\mathsf{M}_s$, a 4-layer MLP is employed, and the reflection direction $\boldsymbol{\omega}_r$ is also positional-encoded using 4 frequency scales. In the first stage, we exclusively focus on decomposing the highlight (largely white) areas. To reduce the complexity of considering color, we assume that the specular radiance is in grayscale and only consider changes in brightness. We can incorporate color information in later stages to obtain a more detailed specular reflection model.

In Stage 2, the light visibility network $\mathsf{M}_\nu$ has 4 layers. To better encode the input 3D coordinate $\boldsymbol{x}$, positional encoding with 10 frequency scales is utilized. The input view direction $\omega_i$ is also positional-encoded using 4 scales. The indirect light network $\mathsf{M}_{\mathrm{ind}}$ in stage 2 comprises 4 layers.

Table 4: Overview of the capabilities of the recent inverse rendering methods.

| Method | Explicit surface extraction | Diffuse/specular color decomposition | Illumination reconstruction | Materials reconstruction | Handles glossy surfaces | No Extra Data? | Code available | Venue |
|---|---|---|---|---|---|---|---|---|
| NeRV Srinivasan et al. (2021) | ✗ | ✓ | ✓ | ✓ | ✗ | ✓ | [to appear] | CVPR 2021 |
| PhySG Zhang et al. (2021a) | ✓ | ✓ | ✓ | ✓ | ✓ | object masks | ✓ | CVPR 2021 |
| NeuS Wang et al. (2021) | ✓ | ✗ | ✗ | ✗ | ✗ | ✓ | ✓ | NeurIPS 2021 |
| NeRFactor Zhang et al. (2021b) | ✗ | ✓ | ✓ | ✓ | ✗ | BRDF dataset | ✓ | SG Asia 2021 |
| Ref-NeRF Verbin et al. (2022) | ✗ | ✓ | ✗ | ✗ | ✓ | ✓ | ✓ | CVPR 2022 |
| NVDiffrec Munkberg et al. (2022) | ✓ | ✓ | ✓ | ✓ | ✗ | object masks | ✓ | CVPR 2022 |
| IndiSG Zhang et al. (2022b) | ✓ | ✓ | ✓ | ✓ | ✗ | object masks | ✓ | CVPR 2022 |
| Geo-NeuS Fu et al. (2022) | ✓ | ✓ | ✗ | ✗ | ✗ | point clouds | ✓ | NeurIPS 2022 |
| BakedSDF Yariv et al. (2023) | ✓ | ✓ | ✗ | ✗ | ✗ | ✓ | ✗ | SG 2023 |
| TensoIR Jin et al. (2023) | ✗ | ✓ | ✓ | ✓ | ✗ | ✓ | ✓ | CVPR 2023 |
| NeFII Wu et al. (2023a) | ✓ | ✓ | ✓ | ✓ | ✗ | ✓ | ✗ | CVPR 2023 |
| Ref-NeuS Ge et al. (2023) | ✓ | ✗ | ✗ | ✗ | ✓ | ✓ | [to appear] | arXiv 2023/03 |
| αSurf Wu et al. (2023b) | ✓ | ✗ | ✗ | ✗ | ✗ | object masks | [to appear] | arXiv 2023/03 |
| NeILF++ Zhang et al. (2023a) | ✓ | ✓ | ✓ | ✓ | ✗ | ✓ | [to appear] | arXiv 2023/03 |
| ENVIDR Liang et al. (2023) | ✓ | ✓ | ✓ | ✓ | ✓ | BRDF dataset | [to appear] | arXiv 2023/03 |
| NeMF Zhang et al. (2023b) | ✗ | ✓ | ✓ | ✓ | ✗ | ✓ | ✗ | arXiv 2023/04 |
| NeAI Zhuang et al. (2023) | ✗ | ✓ | ✓ | ✓ | ✓ | ✓ | [to appear] | arXiv 2023/04 |
| Factored-NeuS(ours) | ✓ | ✓ | ✓ | ✓ | ✓ | ✓ | [to appear] | - |

In stage 3, the encoder part of the BRDF network consists of 4 layers, and the input 3D coordinate is positional-encoded using 10 scales. The output latent vector $z$ has 32 dimensions, and we impose a sparsity constraint on the latent code $z$, following IndiSG (Zhang et al., 2022b). The decoder part of the BRDF network is a 2-layer MLP with a width of 128, and the output has 4 dimensions, including the diffuse albedo $d_a \in \mathbb{R}^3$ and roughness $r \in \mathbb{R}$. Finally, the specular albedo network $\mathsf{M}_{sa}$ uses a 4-layer MLP, where the input 3D coordinate $x$ is positional-encoded using 10 scales, and the input reflection direction $\omega_r$ is positional-encoded using 4 scales.

The learning rate for all three stages begins with a linear warm-up from 0 to $5 \times 10^{-4}$ during the first 5K iterations. It is controlled by the cosine decay schedule until it reaches the minimum learning rate of $2.5 \times 10^{-5}$, which is similar to NeuS. The weights $\lambda_{\text{sur}}$ for the surface color loss are set for 0.1, 0.6, 0.6, and 0.01 for DTU, SK3D, Shiny, and the IndiSG dataset, respectively. We train our model for 300K iterations in the first stage, which takes 11 hours in total. For the second and third stages, we train for 40K iterations, taking around 1 hour each. The training was performed on a single NVIDIA RTX 4090 GPU.

## C  Training strategies of stage 1

In our training process, we define three loss functions, namely volume radiance loss $\mathcal{L}_{\text{vol}}$, surface radiance loss $\mathcal{L}_{\text{sur}}$, and regularization loss $\mathcal{L}_{\text{reg}}$. The volume radiance loss $\mathcal{L}_{\text{vol}}$ is measured by calculating the $\mathcal{L}_1$ distance between the ground truth colors $C^{\text{gt}}$ and the volume radiances $C^{\text{vol}}$ of a subset of rays $\mathcal{R}$, which is defined as follows.

$$\mathcal{L}_{\text{vol}} = \frac{1}{|\mathcal{R}|} \sum_{r \in \mathcal{R}} \|C_r^{\text{vol}} - C_r^{\text{gt}}\|_1 \tag{17}$$

The surface radiance loss $\mathcal{L}_{\text{sur}}$ is measured by calculating the $\mathcal{L}_1$ distance between the ground truth colors $C^{\text{gt}}$ and the surface radiances $C^{\text{sur}}$. During the training process, only a few rays have intersection points with the surface. We only care about the set of selected rays $\mathcal{R}'$, which satisfies the condition that each ray exists point whose SDF value is less than zero and not the first sampled point. The loss is defined as follows.

$$\mathcal{L}_{\text{sur}} = \frac{1}{|\mathcal{R}'|} \sum_{r \in \mathcal{R}'} \|C_r^{\text{sur}} - C_r^{\text{gt}}\|_1 \tag{18}$$

$\mathcal{L}_{\text{reg}}$ is an Eikonal loss term on the sampled points. Eikonal loss is a regularization loss applied to a set of sampling points $X$, which is used to constrain the noise in signed distance function (SDF) generation.

$$\mathcal{L}_{\text{reg}} = \frac{1}{|\mathcal{X}|} \sum_{x \in \mathcal{X}} (\|\nabla \mathsf{S}_\theta(x)\|_2 - 1)^2 \tag{19}$$

We use weights $\lambda_{\text{sur}}$ and $\lambda_{\text{reg}}$ to balance the impact of these three losses. The overall training weights are as follows.

$$\mathcal{L} = \mathcal{L}_{\text{vol}} + \lambda_{\text{sur}}\mathcal{L}_{\text{sur}} + \lambda_{\text{reg}}\mathcal{L}_{\text{reg}} \tag{20}$$

## D  Details of stage 2

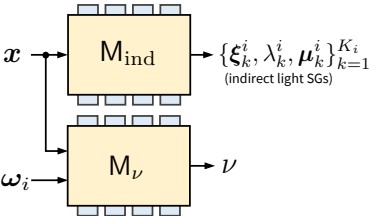

Figure 6: Overview for Stage 2. $x$ is a point on the surface. $\omega_i$ is the view direction. Indirect light and light visibility network $M_{\text{ind}}$ and $M_v$ produce their respective indirect light SGs and light visibility $v$.

At this stage, we focus on predicting the lighting visibility and indirect illumination of a surface point $\boldsymbol{x}$ under different incoming light direction $\boldsymbol{\omega}_i$ using the SDF in the first stage. Therefore, we need first to calculate the position of the surface point $\boldsymbol{x}$. In stage one, we have calculated two sampling points $\boldsymbol{r}(t_{i'-1}), \boldsymbol{r}(t'_i)$ near the surface. As Geo-NeuS (Fu et al., 2022), we weigh these two sampling points to obtain a surface point $\boldsymbol{x}$ as follows.

$$\boldsymbol{x} = \frac{\mathsf{S}_\theta(\boldsymbol{r}(t_{i'-1}))\boldsymbol{r}(t'_i) - \mathsf{S}_\theta(\boldsymbol{r}(t'_i))\boldsymbol{r}(t_{i'-1})}{\mathsf{S}_\theta(\boldsymbol{r}(t_{i'-1})) - \mathsf{S}_\theta(\boldsymbol{r}(t'_i))} \tag{21}$$

**Learning lighting visibility.** Visibility is an important factor in shadow computation. It calculates the visibility of the current surface point $\boldsymbol{x}$ in the direction of the incoming light $\boldsymbol{\omega}_i$. Path tracing of the SDF is commonly used to obtain a binary visibility (0 or 1) as used in IndiSG (Zhang et al., 2022b), but this kind of visibility is not friendly to network learning. Inspired by NeRFactor (Zhang et al., 2021b), we propose to use an integral representation with the continuous weight function $w(t)$ (from 0 to 1) for the SDF to express light visibility. Specifically, we establish a neural network $\mathsf{M}_\nu : (\boldsymbol{x}, \boldsymbol{\omega}_i) \mapsto \nu$, that maps the surface point $\boldsymbol{x}$ and incoming light direction $\boldsymbol{\omega}_i$ to visibility, and the ground truth value of light visibility is obtained by integrating the weights $w_i$ of the SDF of sampling points along the incoming light direction and can be expressed as follows.

$$\nu^{gt} = 1 - \sum_{i=1}^{n} w_i \tag{22}$$

The weights of the light visibility network are optimized by minimizing the loss between the calculated ground truth values and the predicted values of a set of sampled incoming light directions $\Omega_i \subset \mathbb{S}^2$. This pre-integrated technique can reduce the computational burden caused by the integration for subsequent training.

$$\mathcal{L}_{\text{vis}} = \frac{1}{|\Omega_i|} \sum_{\boldsymbol{\omega} \in \Omega_i} \|\nu_{\boldsymbol{\omega}} - \nu_{\boldsymbol{\omega}}^{\text{gt}}\|_1 \tag{23}$$

**Learning indirect illumination.** Indirect illumination refers to the light that is reflected or emitted from surfaces in a scene and then illuminates other surfaces, rather than directly coming from a light source, which contributes to the realism of rendered images. Following IndiSG (Zhang et al., 2022b), we parameterize indirect illumination $I(\boldsymbol{x}, \boldsymbol{\omega}_i)$ via $K_i = 24$ Spherical Gaussians (SGs) as follows.

$$I(\boldsymbol{x}, \boldsymbol{\omega}_i) = \sum_{k=1}^{K_i} I_k(\boldsymbol{\omega}_i \mid \boldsymbol{\xi}_k^i(\boldsymbol{x}), \lambda_k^i(\boldsymbol{x}), \boldsymbol{\mu}_k^i(\boldsymbol{x})) \tag{24}$$

where $\boldsymbol{\xi}_k^i(\boldsymbol{x}) \in \mathbb{S}^2$, $\lambda_k^i(\boldsymbol{x}) \in \mathbb{R}_+$, and $\boldsymbol{\mu}_k^i(\boldsymbol{x}) \in \mathbb{R}^3$ are the lobe axis, sharpness, and amplitude of the $k$-th Spherical Gaussian, respectively. For this, we train a network $\mathsf{M}_{\text{ind}} : \boldsymbol{x} \mapsto \{\boldsymbol{\xi}_k^i, \lambda_k^i, \boldsymbol{\mu}_k^i\}_{k=1}^{K_i}$ that maps the surface point $\boldsymbol{x}$ to the parameters of indirect light SGs. Similar to learning visibility, we randomly sample several directions $\boldsymbol{\omega}_i$ from the surface point $\boldsymbol{x}$ to obtain (pseudo) ground truth $I^{\text{gt}}(\boldsymbol{x}, \boldsymbol{\omega}_i)$. Some of these rays have intersections $\boldsymbol{x}'$ with other surfaces, thus, $\boldsymbol{\omega}_i$ is the direction pointing from $\boldsymbol{x}$ to $\boldsymbol{x}'$. We query our proposed color network $\mathsf{M}_c$ to get the (pseudo) ground truth indirect radiance $I^{\text{gt}}(\boldsymbol{x}, \boldsymbol{\omega}_i)$ as follows.

$$I^{\text{gt}}(\boldsymbol{x}, \boldsymbol{\omega}_i) = \mathsf{M}_c(\boldsymbol{x}', \boldsymbol{n}', \boldsymbol{\omega}_i, \boldsymbol{v}_f) \tag{25}$$

where $\boldsymbol{n}'$ is the normal on the point $\boldsymbol{x}'$. We also use $\mathcal{L}_1$ loss to train the network.

$$\mathcal{L}_{\text{ind}} = \frac{1}{|M|} \sum_{m \in M} \|I(\boldsymbol{x}, \boldsymbol{\omega}_m) - I_m^{\text{gt}}(\boldsymbol{x}, \boldsymbol{\omega}_m)\|_1 \tag{26}$$

## E    DETAILS OF STAGE 3

The combination of light visibility and illumination SG is achieved by applying a ratio to the lobe amplitude of the output SG, while preserving the center position of the SG. We randomly sample $K_s = 32$ directions within the SG lobe and compute a weighted average of the visibility with different directions.

$$\nu(\boldsymbol{x}, \boldsymbol{\omega}_i) \otimes E_k(\boldsymbol{\omega}_i \mid \boldsymbol{\xi}_k^e, \lambda_k^e, \boldsymbol{\mu}_k^e) \approx E_k(\boldsymbol{\omega}_i \mid \boldsymbol{\xi}_k^e, \lambda_k^e, \frac{\sum_{s=1}^{K_s} E_k(\boldsymbol{\omega}_s)\nu(\boldsymbol{x}, \boldsymbol{\omega}_s)}{\sum_{s=1}^{K_s} E_k(\boldsymbol{\omega}_s)} \boldsymbol{\mu}_k^e) \tag{27}$$

Here, we offer intuitive explanations for why the incorporation of specular albedo in the model results in a decrease in lighting prediction. The increase in the model's complexity is the primary reason. Specular albedo introduces a more detailed modeling of surface reflection characteristics, requiring additional parameters and learning capacity. This raises the difficulty of training the model, potentially resulting in overfitting or training instability, thereby affecting the accurate prediction of lighting.

## F  ADDITIONAL RESULTS

We show the qualitative results for surface reconstruction compared with NeuS, Ref-NeRF, IndiSG, PhySG, and NVDiffrec on the Shiny dataset, which is a synthetic dataset with glossy surfaces. From Fig. 7, we can observe that NeuS is easily affected by highlights, and the geometry reconstructed by Ref-NeRF has strong noise. PhySG is slightly better than IndiSG on the Shiny synthetic dataset with jointly optimizing materials and surfaces, such as toaster and car scenes, but still can not handle complex highlights. NVDiffrec works well on the teapot model but fails on other more challenging glossy surfaces. Our method is able to produce clean glossy surfaces without being affected by the issues caused by highlights. Overall, our approach demonstrates superior performance in surface reconstruction, especially on glossy surfaces with specular highlights.

In order to have a fair comparison with Geo-NeuS on the DTU dataset, we incorporate the components of Geo-NeuS based on the additional data (the point clouds from SfM and image pairs) used in Geo-NeuS into our method. As shown in Tab. 5, our approach can further enhance the surface reconstruction quality on datasets where highlights are less pronounced.

Table 5: Quantitative results in terms of Chamfer distance on DTU (Jensen et al., 2014).

|  | DTU 63 | DTU 97 | DTU 110 | **Mean** |
|---|---|---|---|---|
| Geo-NeuS Fu et al. (2022) | 0.96 | 0.91 | 0.70 | 0.86 |
| Factored-NeuS (ours) | 0.99 | 1.15 | 0.89 | 1.01 |
| Factored-NeuS (ours w/ Geo) | **0.95** | **0.89** | **0.69** | **0.84** |

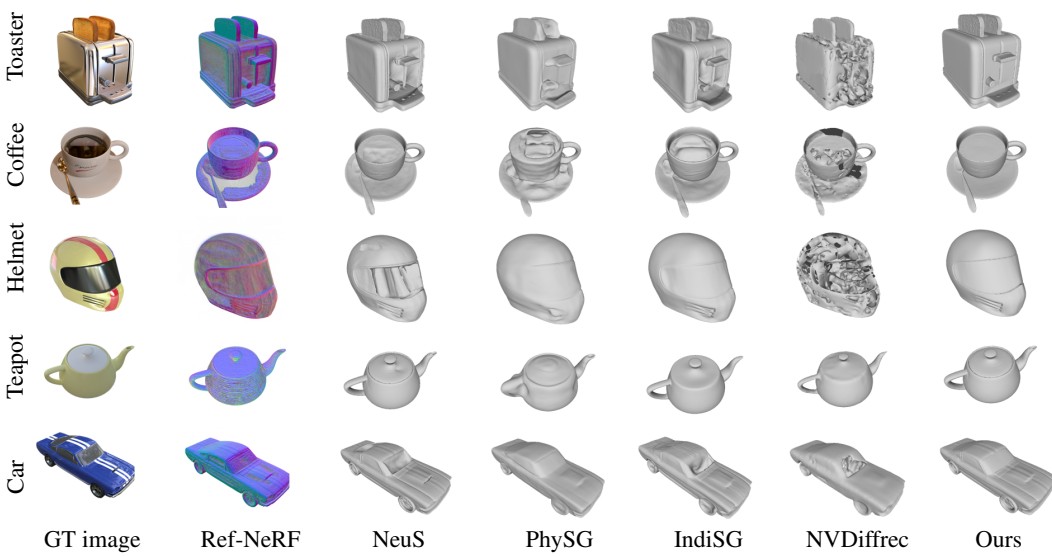

Figure 7: Qualitative results for the Shiny dataset (Verbin et al., 2022).

In addition to the synthetic datasets with ground truth decomposed materials, we also provide qualitative results on real-world captured datasets such as DTU and SK3D in Fig. 8. From the DTU data, we can observe that our method has the ability to separate the specular reflection component from the diffuse reflection component, as seen in the highlights on the apple, can, and golden rabbit. Even when faced with a higher intensity of specular reflection, as demonstrated in the example showcased in SK3D, our method excels at preserving the original color in the diffuse part and accurately separating highlights into the specular part.

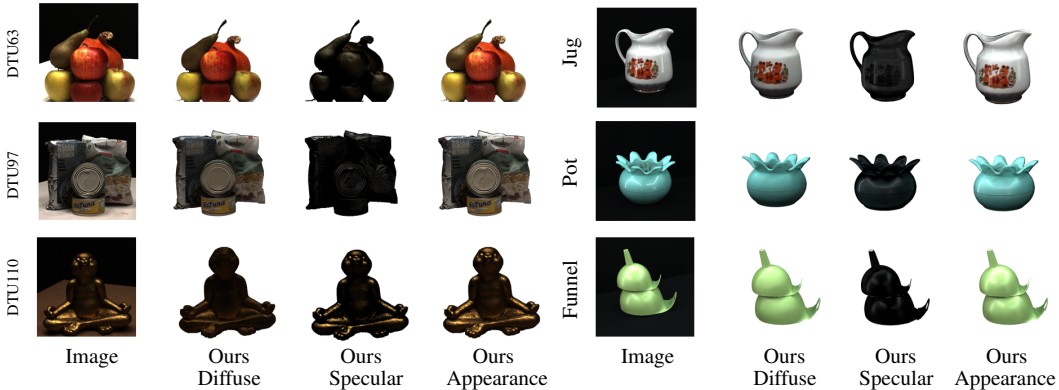

Figure 8: Qualitative results for the DTU (left) and SK3D (right) datasets.

We conduct another experiment to compare our modeling approach with Ref-NeRF and $S^3$-NeRF (Yang et al., 2022). The experimental quantitative and qualitative results are shown in Fig. 11. Ref-NeRF utilizes volume rendering colors for diffuse and specular components. If we directly combine SDF and the architecture of Ref-NeRF, it is challenging to eliminate the influence of highlights. Furthermore, if we applied the construction method of $S^3$-NeRF, which involves integrating surface rendering colors into volume rendering, to modify our model structure, we found that this modeling approach cannot address the issue of geometric concavity caused by highlights.

We visualize the decomposition of diffuse and specular in the first stage in Fig. 12. In the first stage, the decomposition of diffuse and specular is not a true BRDF model. This is because the MLP in the first stage is used solely for predicting the components of diffuse and specular reflection, rather than predicting material properties such as albedo and roughness. The decision to directly predict colors instead of material properties in the first stage serves two purposes: reducing model complexity by focusing on the direct prediction of specular reflection color, and optimizing geometry for better reconstruction. By decomposing highlights through the network in the first stage, surfaces with specular reflections can be reconstructed more effectively, demonstrated by the presence of flower pot ablation, and without encountering the concavity issues observed in other methods.

Additionally, in Fig. 13,Fig. 14, and Fig. 15, we presented the rendering, albedo, roughness, diffuse color, specular color, light visibility, indirect light, environment light results for the IndiSG, DTU and SK3D datasets, respectively.

Table 6: Ablation study of indirect light

| | Baloons | | | Hotdog | | | Chair | | | Jugs | | |
| --- | --- | --- | --- | --- | --- | --- | --- | --- | --- | --- | --- | --- |
| | albedo | illumination | rendering | albedo | illumination | rendering | albedo | illumination | rendering | albedo | illumination | rendering |
| w/ Lvis w/o IndiLgt | 23.13 | 18.24 | 29.45 | 25.62 | 17.97 | 35.97 | 25.22 | 18.04 | 34.31 | 22.87 | 21.84 | 26.30 |
| Ours | **25.79** | **21.79** | **33.89** | **30.72** | **20.23** | **36.71** | **26.33** | **20.97** | **34.58** | **22.94** | **21.84** | **36.48** |

In stage 3, if we do not consider indirect illumination during the training process, the predicted results for rendering, material, and lighting will all experience a decline. The results are shown in Fig. 16. The specific PSNR metrics can be found in Tab. 6

NeRO (Liu et al., 2023) is capable of reconstructing surfaces with highlights effectively. We further conducted additional comparisons of NeRO and NeuS on datasets DTU, SK3D, and Glossy.

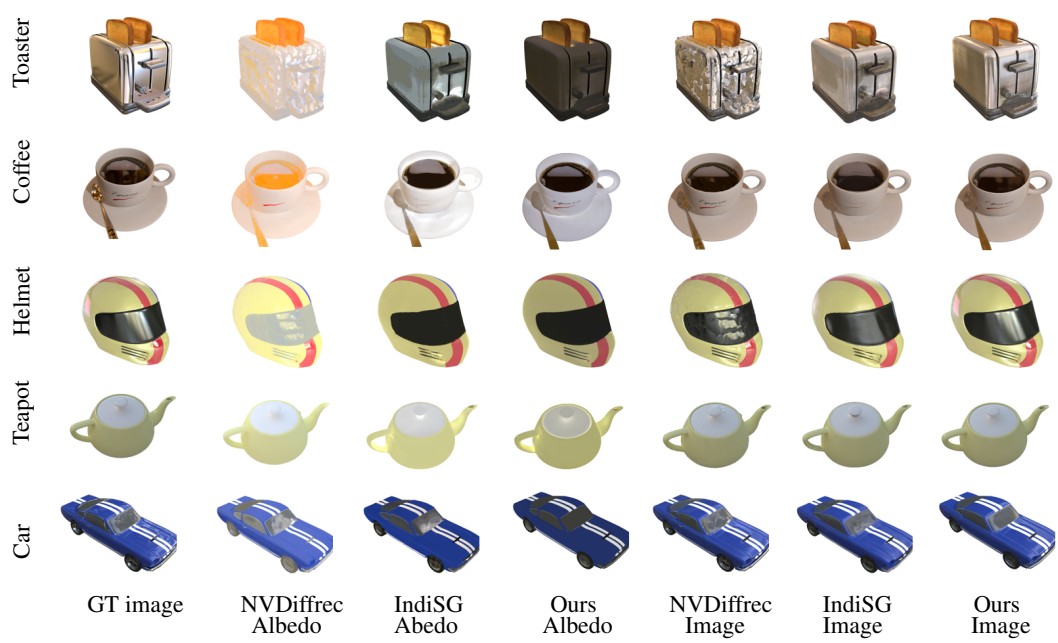

Figure 9: Qualitative results for the Shiny dataset. Albedo refers to the diffuse albedo.

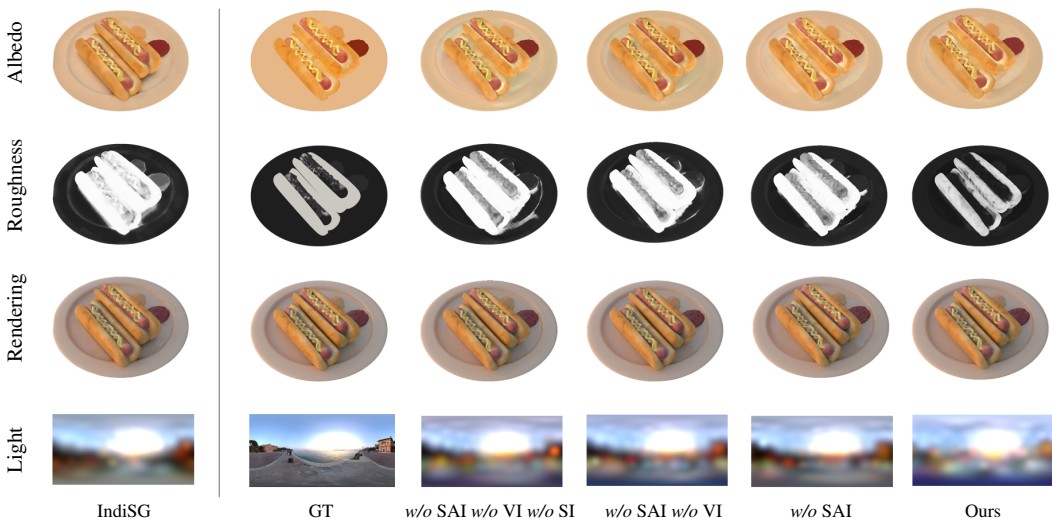

Figure 10: Ablation study of material and illumination reconstruction.

For DTU datasets, as shown in Fig. 17 NeRO not only struggles to accurately restore detailed information but also fails to address the negative impact of partial highlights on geometry. Moreover, the presence of shadows causes NeRO to mistakenly reconstruct shadowed areas as real objects and fill them in(bricks and skull models). Additional Chamfer distance metrics for the DTU dataset are presented in Tab. 7.

As shown in Fig. 18, when running the SK3D dataset, similar issues were encountered as in the DTU dataset. Even in scenarios with a simple background and straightforward geometry, NeRO still tends

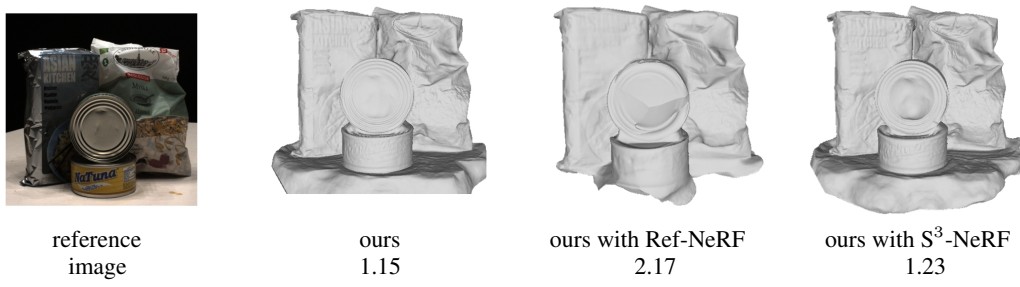

Figure 11: Comparison with Ref-NeRF and $S^3$-NeRF.

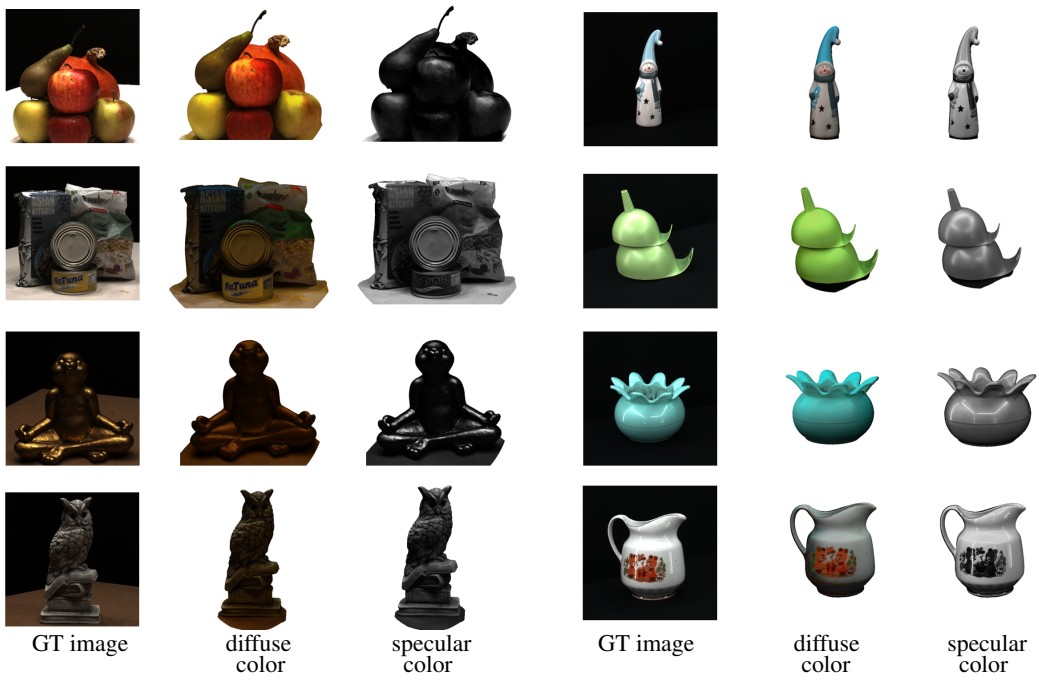

Figure 12: Diffuse and specular decomposition results in the first stage.

to lose certain details of the dataset and fills in shadowed areas. The Chamfer distance metrics results are presented in Tab. 8

On the Glossy dataset, NeRO performs better, but our method is also capable of mitigating the impact of highlights on geometry as shown in Fig. 19. Moreover, compared to NeuS, the results show a significant improvement. Additional Chamfer distance metrics for the glossy dataset are presented in Tab. 9. The Glossy real datasets include bear, bunny, coral, vase. The rest of the others are in the glossy synthetic dataset.

In Fig. 20, we additionally showcase some visualization results of relighting with IndiSG. IndiSG and ours yield different predictions for material, resulting in variations in the relighting results, but the relighting results generated by our method exhibit richer details. Our method demonstrates the practical utility employed in the relighting scenarios.

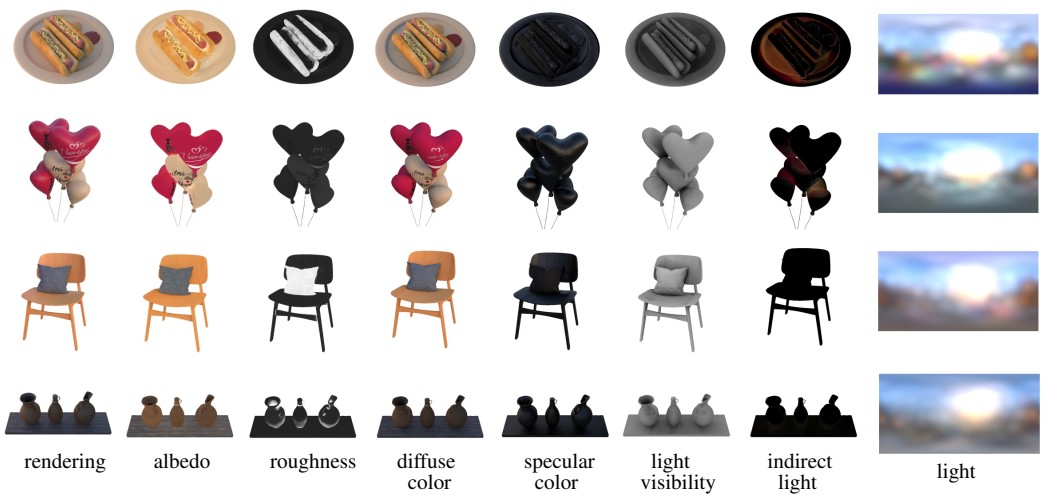

rendering    albedo    roughness    diffuse color    specular color    light visibility    indirect light    light

Figure 13: Visualization of all components on IndiSG dataset.

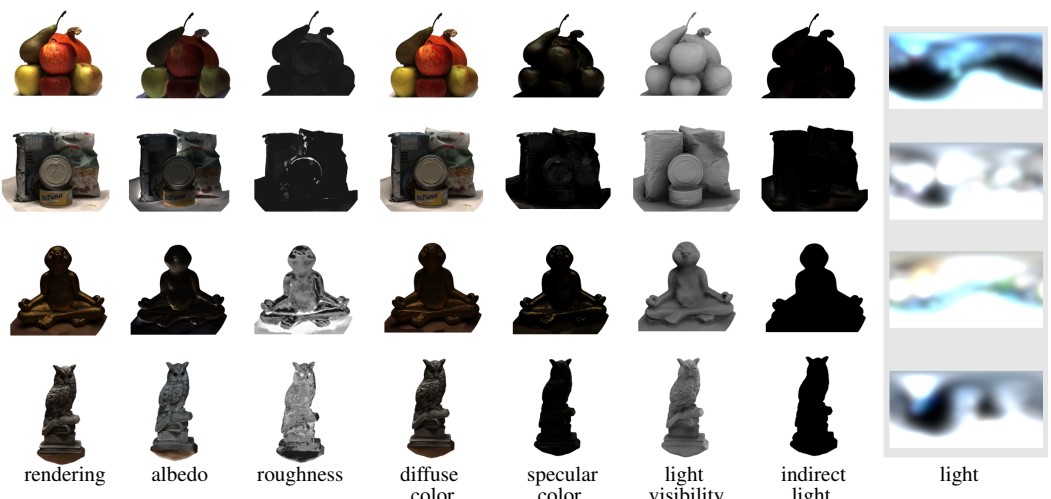

rendering    albedo    roughness    diffuse color    specular color    light visibility    indirect light    light

Figure 14: Visualization of all components on DTU dataset.

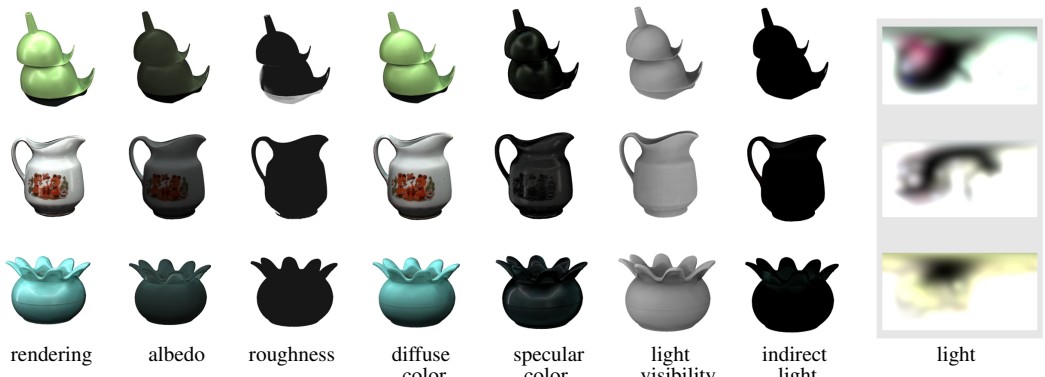

rendering    albedo    roughness    diffuse color    specular color    light visibility    indirect light    light

Figure 15: Visualization of all components on SK3D dataset.

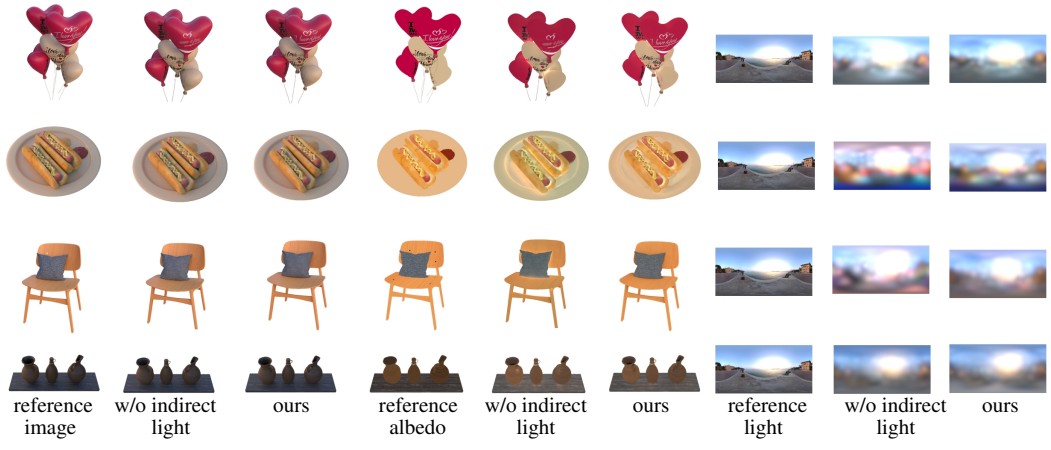

Figure 16: Ablation study of indirect light.

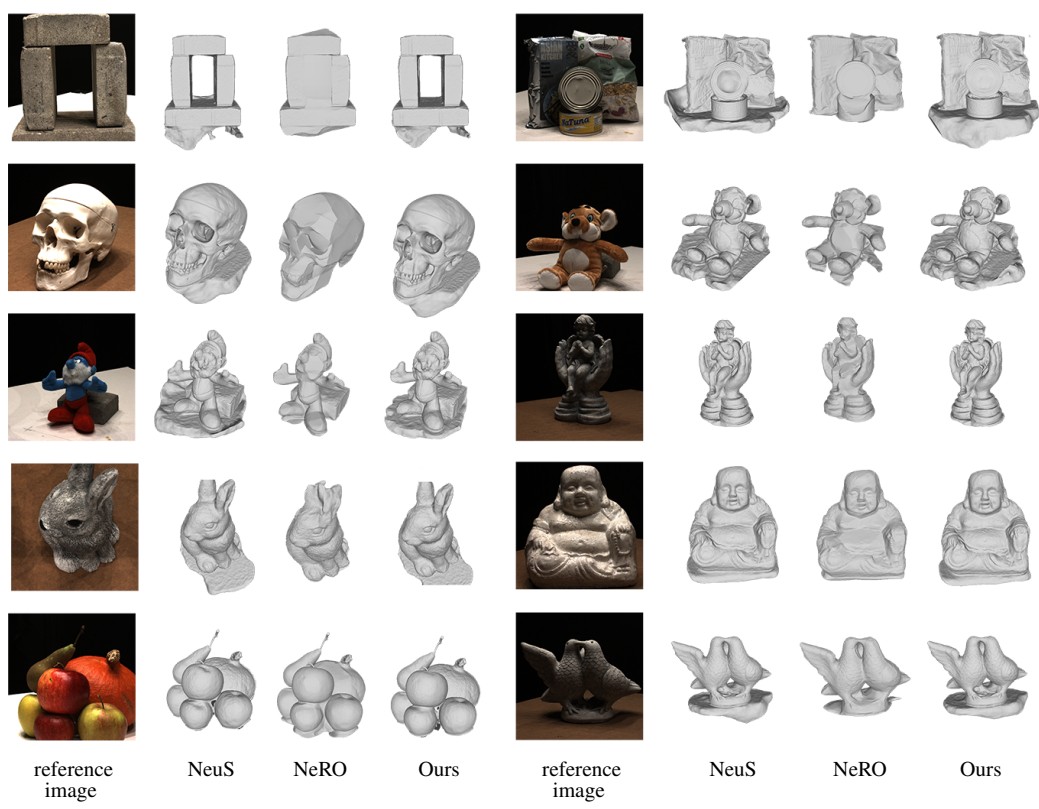

Figure 17: Comparison with NeRO and NeuS on DTU dataset.

Table 7: Comparison with NeRO and NeuS on DTU dataset.

| DTU | 24 | 37 | 40 | 55 | 63 | 65 | 83 | 97 | 105 | 106 | 110 | 114 | 118 | 122 | mean |
|------|------|------|------|------|------|------|------|------|------|------|------|------|------|------|------|
| NeuS | 1.00 | 1.37 | 0.93 | 0.43 | 1.01 | 0.65 | 1.48 | 1.21 | 0.83 | **0.52** | 1.14 | **0.35** | 0.49 | 0.54 | 0.85 |
| NeRO | 1.10 | 1.13 | 1.26 | 0.46 | 1.32 | 1.93 | 1.61 | 1.47 | 1.10 | 0.70 | 1.14 | 0.39 | 0.52 | 0.57 | 1.05 |
| Ours | **0.82** | **1.05** | **0.85** | **0.40** | **0.99** | **0.59** | **1.44** | **1.15** | **0.81** | 0.58 | **0.89** | 0.36 | **0.44** | **0.46** | **0.77** |

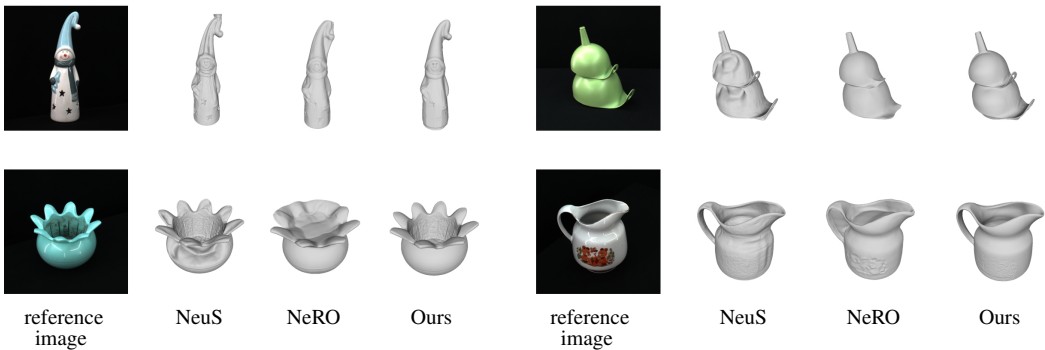

| reference image | NeuS | NeRO | Ours | reference image | NeuS | NeRO | Ours |

Figure 18: Comparison with NeRO and NeuS on SK3D dataset.

Table 8: Comparison with NeRO and NeuS on SK3D dataset.

| SK3D | NeuS | NeRO | Ours |
|---|---|---|---|
| pot | 2.09 | 6.03 | **1.54** |
| jug | 1.81 | 4.23 | **1.40** |
| funnel | 3.93 | 2.63 | **1.95** |
| snowman | 1.40 | 1.71 | **1.31** |
| mean | 2.31 | 3.65 | **1.55** |

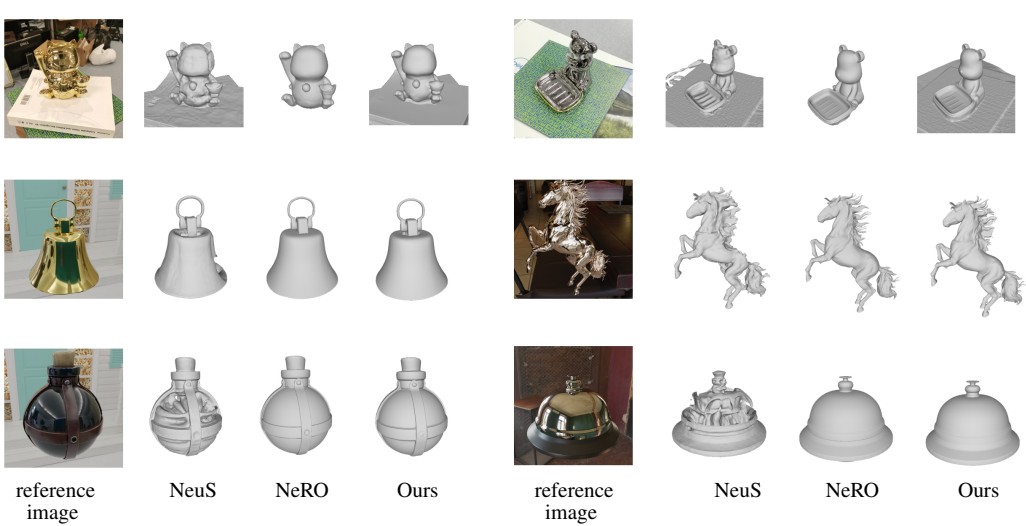

| reference image | NeuS | NeRO | Ours | reference image | NeuS | NeRO | Ours |

Figure 19: Comparison with NeRO and NeuS on Glossy dataset.

Table 9: Comparison with NeRO and NeuS on Glossy dataset.

| Glossy | bear | bunny | coral | maneki | vase | angel | bell | cat | horse | luyu | potion | tbell | teapot | mean |
|---|---|---|---|---|---|---|---|---|---|---|---|---|---|---|
| NeuS | 0.0074 | 0.0022 | 0.0016 | 0.0091 | 0.0101 | 0.0035 | 0.0146 | 0.0278 | 0.0053 | 0.0066 | 0.0393 | 0.0348 | 0.0546 | 0.0167 |
| NeRO | **0.0033** | **0.0012** | **0.0014** | **0.0024** | **0.0011** | **0.0034** | **0.0032** | **0.0044** | **0.0049** | **0.0054** | **0.0053** | **0.0035** | **0.0037** | **0.0033** |
| Ours | 0.0034 | 0.0017 | **0.0014** | 0.0027 | 0.0023 | **0.0034** | 0.0054 | 0.0059 | 0.0052 | 0.0060 | 0.0058 | **0.0035** | 0.0105 | 0.0044 |

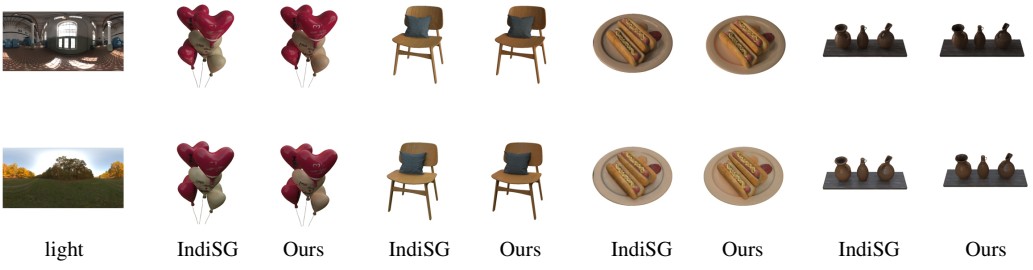

light     IndiSG     Ours     IndiSG     Ours     IndiSG     Ours     IndiSG     Ours

Figure 20: Relighting comparison with IndiSG.

