# OpenReview forum: "Factored-NeuS: Reconstructing Surfaces, Illumination, and Materials of Possibly Glossy Objects"
_ICLR.cc/2024/Conference — Submitted to ICLR 2024_

### Official Review · Reviewer_8TvC · 2023-10-30

**Soundness:** 3 good
**Presentation:** 4 excellent
**Contribution:** 3 good
**Rating:** 6
**Confidence:** 5

**Summary:**

This paper develops a method that recovers the surface geometry, materials (diffuse and specular), and illumination of a scene from multi-view images. Unlike previous approaches, this method does not require any extra data and is adept at managing glossy objects and high intensity lighting conditions.

The method has three stages: In the first stage, the scene's radiance and Signed Distance Function (SDF) are reconstructed. A novel training strategy is introduced here, which leverages both volume and surface rendering. This strategy effectively addresses complex, view-dependent lighting effects during surface reconstruction. In the second stage, light visibility and indirect illumination are distilled from the learned SDF and radiance field using learnable neural networks. Finally, materials and direct illuminations are recovered using a set of Spherical Gaussians.

**Strengths:**

- A significant focus of the paper is on addressing the challenges posed by glossy scenes, especially when they feature specular highlights. This is an important area as many objects in the real world manifest specular and many previous method fail to achieve good quality it in these areas.
- The paper is well-written. It clearly outlines previous works and offers a balanced comparison with them. The paper's contributions are clearly stated, and it provides a comprehensive ablation study on its proposed enhancements.
- From the experiments, it's evident that this paper achieves state-of-the-art (SOTA) performance in terms of material and geometry reconstruction. The ablation study further reinforces the validity of the improvements proposed.
- I find the first stage of this paper particularly innovative. It was previously unknown that optimizing both volume and surface rendering concurrently could enhance surface geometry, especially in specular regions. This concept might be applied in other contexts to address surface geometry challenges arising from pronounced specular highlights.

**Weaknesses:**

- This paper appears to be a combination of numerous prior works. From my perspective, the first stage borrows from the papers NeuS and PhySG. The second stage seems inspired by NeRFactor and IndiSG, particularly in how they learn visibility mapping from a previously trained geometry neural network. The final stage also bears similarities to IndiSG. Given these observations, I contend that the paper presents limited novelty in its three-stage training approach.
- The proposed Specular Albedo Index (SAI) lacks solid theoretical grounding. The authors don't clearly explain why the specular albedo is applied only to the indirect component (as seen in eq-13) and not to the direct component (eq-10). Furthermore, the experiments indicate that the introduction of SAI causes the method to underperform in its illumination estimation. It would be beneficial if the authors could elaborate on this performance decline and offer deeper analysis and insights

**Questions:**

Please address the question about SAI in the weakness part.

In summary, I believe this paper offers a robust solution for determining the geometry, material, and illumination of objects in uncontrolled settings using only multi-view inputs. The authors outline a three-stage approach to address this complex problem and provide valuable insights. However, certain concepts and design choices remain ambiguous. It would be beneficial if the authors could address these in their response.

---

> ### Author Response · Authors · 2023-11-23
> **Authors response**
>
> We sincerely appreciate the thoughtful comments provided by the reviewers and we respond to each question in the following.
>
> >[W1]  This paper appears to be a combination of numerous prior works.
>
>
> Our first stage is fundamentally different from other work in both the implementation and the design concept behind it. The other two stages build on previous work, but they are modifications, to improve the results and therefore also contain novelty. Also, the way our system is constructed is novel. We do use and build upon components from previous work, but creating a complete and well working system and adapting existing components and combining them with our novel Stage 1 is still a significant challenge.  We provide more details below.
>
> >[W1 / 1] The first stage borrows from the papers NeuS and PhySG.
>
> In the first stage, we proposed a novel surface reconstruction method. The volume rendering module is better suited to radiance reconstruction and the surface rendering module ensures higher quality specular and diffuse separation close to the surface. Both working together enable accurate and efficient reconstruction of glossy object surfaces. Previous methods consider only using volume rendering (NeuS, VolSDF) or surface rendering IDR(PhySG, IndiSG) to reconstruct geometric information, and cannot handle the reconstruction of glossy objects, as shown in Tab. 1, Fig. 3, and Appx Fig. 7.
>
> >[W1 / 2]  The second stage seems inspired by NeRFactor and IndiSG
>
> We have indeed incorporated the concept of continuous light visibility from NeRFactor to enhance the binary light visibility of IndiSG, which predicts light visibility as either 0 or 1. We predict light visibility as 1-Σw. Existing ablation studies demonstrate that employing this approach yields better results in Tab 3 and Fig 10. Indirect lighting is currently a widely used method for lighting modeling, and we do not claim novelty in this aspect.
>
> >[W1 / 3]  The final stage also bears similarities to IndiSG
>
> In the third stages, we present a specular albedo estimation method based on spherical Gaussians. Although the component is simple, the exceptional effectiveness of our way of utilizing them is evident in Tab. 2, Fig. 4, and Appx Fig. 10.
>
> >[W2]  The proposed Specular Albedo Index (SAI) lacks solid theoretical grounding. The authors don't clearly explain why the specular albedo is applied only to the indirect component (as seen in eq-13) and not to the direct component (eq-10).
>
> Both direct and indirect lighting incorporate the specular albedo, as described in Eq. 11.
>
> >[W3]  Furthermore, the experiments indicate that the introduction of SAI causes the method to underperform in its illumination estimation. It would be beneficial if the authors could elaborate on this performance decline and offer deeper analysis and insights
>
> When our model complexity is more inclined towards focusing on materials, it restricts the complexity of the lighting model, leading to a decline in the quality of lighting reconstruction. We have added this analysis in the section E of appendix.

---

### Official Review · Reviewer_nhZb · 2023-11-01

**Soundness:** 4 excellent
**Presentation:** 3 good
**Contribution:** 4 excellent
**Rating:** 8
**Confidence:** 4

**Summary:**

This paper describes a method for building a model of scene radiances out of a set of images of the object from many orientations.  The model factors the object surface radiance into four parts: ambient body reflection, ambient surface reflection, direct body reflection, and direct surface reflection. The authors use multiple trained networks to build the model, divided into three stages. The new method is compared against three other recently published methods that also build decomposed NeRF models.

**Strengths:**

The strength of the paper is the method of decomposition, which follows more closely with prior work on physical models of reflection than comparable methods (even if the authors didn't realize it).  The use of multiple networks to model different phenomena also factors the problem into more learnable parts.

A second strength is the focus on shape as a way to build ground truth into the decomposition.  Developing good ground truth for body and surface reflection is challenging at best for real imagery.  However, by focusing the loss on surface reconstruction, which is a derivative of the estimate of body and surface reflection, they were able to provide reasonable feedback to the decomposition networks.

I like the fact that the networks must be learning linear color functions (which is necessary because adding body and surface reflection in sRGB space would produce the wrong values).

The qualitative and quantitative results are strong for the data sets evaluated.  I also like the use of real data sets rather than relying on synthetic ones.

**Weaknesses:**

The primary weakness is a lack of consistency with vocabulary and definitions.  There is a long history of physical models of appearance, and the model the authors derive is effectively the Bi-illuminant Dichromatic Reflection model (see equations 9 and 10 of Maxwell et. al, CVPR 2008) which is built on the Dichromatic Reflection model (Shafer, 1986).  The physics-based vision community has long used the terms body reflection and surface reflection to refer to what the authors are calling diffuse radiance and specular radiance.  Using a phrase like "surface diffuse radiance" is mixing this vocabulary and creates confusion.

Another confusing example is the sentence  "Diffuse radiance refers to the scattered light that illuminates a surface or space evenly and without distinct shadows or reflections."   Illumination is irradiance onto a surface, while radiance is light leaving the surface.  I believe what the authors are trying to describe is what is often referred to as ambient illumination, which is the light arriving at a surface from all directions except those that point at a direct light source (the source of shadows and shading).  The ambient illumination is then what is reflected by the surface to create ambient body reflection (diffuse diffuse radiance?) and ambient surface reflection (diffuse specular radiance ?).  Using body reflection and surface reflection makes the concepts more clear because it refers to the mechanism of reflection rather than a descriptive term.

The term "diffuse albedo" is also somewhat confusing, though albedo is commonly used (for greyscale?).  Using something like body reflection color is less confusing.

My only other complaint is the reference to a number of appendices, which aren't present in the review copy.  Is it possible to squeeze just the critical points into the actual paper?

**Questions:**

Are the authors using linear or sRGB data to train the networks?  If linear, then is the original data linear, or are they taking the sRGB inverse of sRGB JPG data?

Just to clarify, can the stage 3 models of direct illumination learn multiple direct illuminants?  How is this related to the visibility calculations in stage 2?  Or is there an implicit assumption of a single direct illuminant?

Are there any examples of multiple direct illuminants in the data sets evaluated?

Following Bonneel et. al, 2017, a useful evaluation of intrinsic decomposition is to demonstrate that the decompositions are good enough to enable editing of illumination or reflectance separately. Is there a similar task whereby the authors could show a practical use of the decomposition and use it for comparison with other methods?

---

> ### Author Response · Authors · 2023-11-23
> **Authors response (part 1/2)**
>
> We sincerely appreciate the thoughtful comments provided by the reviewers and we respond to each question in the following.
>
> >[W1]  The lack of consistency with vocabulary and definitions for diffuse radiance and specular radiance.
>
> Thank you for your suggestion. Currently, the terminology is mainly aligned with the 'neural radiance field' used in NeRF for representing colors. Due to your suggestion, we realized that our definition in the main text should be improved. We change the description of diffuse radiance in the main text using “We use the term diffuse radiance to refer to the component of the reflected radiance that stems from a diffuse surface reflection”. We would like to break down the radiance measured by a pixel in the image into two components, the diffuse radiance and the specular radiance. The diffuse radiance is the component due to a diffuse surface reflection and the specular radiance includes all other non-diffuse components.  Regarding the confusion of the usage of the term “surface diffuse radiance”: we simply meant diffuse radiance (as used in other parts of the paper) that stems from the surface rendering part of stage one. We have modified this terminology in the revised manuscript. “We propose to simultaneously optimize the radiance from the volumetric rendering and the surface rendering. For surface rendering, we further split the reflected radiance into a diffuse and a specular component.”
>
> These terms are also used in physically-based rendering and atmospheric science. For example “Advanced Global Illumination”, chapter 2 “The physics of light transport”.
>
> We also checked the reference “A bi-illuminant dichromatic reflection model for understanding images”, but we think the used terminology is not common in the NerF literature. To better align our work with the most relevant previous work, we therefore prefer not to change the terminology. There are many different communities working on reconstruction, lighting, and appearance models and we understand that terminology is not always consistent between communities.
>
> >[W2] The term "diffuse albedo" is also somewhat confusing, though albedo is commonly used (for greyscale?). Using something like body reflection color is less confusing.
>
> Many literatures use diffuse albedo and albedo interchangeably. To distinguish our specular albedo, we opt for diffuse albedo. This terminology is also used by multiple other papers cited by our work. For example, see PhySG, NeRD, and IndiSG.
>
> >[W3] My only other complaint is the reference to a number of appendices, which aren't present in the review copy. Is it possible to squeeze just the critical points into the actual paper?
>
> Thank you for your suggestion. Due to space constraints, it's challenging to incorporate all content into the main text. We have revised the document to place the appendix after the references, ensuring hyperlinks for images and tables to facilitate viewing and review. We also have uploaded a joint revised version to make the review easier.
>
> >[Q1] Are the authors using linear or sRGB data to train the networks? If linear, then is the original data linear, or are they taking the sRGB inverse of sRGB JPG data?
>
> The training data consists of sRGB JPG or PNG images. However, we use  linear data to train the MLP network, which means the output from the MLP, representing diffuse and specular colors, is in linear color space. Subsequently, we apply the linear-to-sRGB equation (Eq 6) to transform the results before utilizing them for training.

---

> ### Author Response · Authors · 2023-11-23
> **Authors response (part 2/2)**
>
> >[Q2]  Just to clarify, can the stage 3 models of direct illumination learn multiple direct illuminants? Are there any examples of multiple direct illuminants in the data sets evaluated?
>
> In the third stage, we use environment maps to represent direct lighting, and environment maps have the capability to represent multiple direct light illuminants. We have conducted an experiment to show the capability to learn multiple direct illuminants in the appendix. We include images showcasing the lighting reconstruction for DTU scans 97 (tin) in Appx Fig.14, observing that the lighting consists of multiple direct light illuminants from the reconstructed environment maps.
>
> >[Q3]  How is the illumination related to the visibility calculations in stage 2? Or is there an implicit assumption of a single direct illuminant?
>
> For the visibility calculations, we demonstrate the method of combining light visibility with spherical Gaussians for illumination in Section  E in the revised appendix.
>
> >[Q4]  Following Bonneel et. al, 2017, a useful evaluation of intrinsic decomposition is to demonstrate that the decompositions are good enough to enable editing of illumination or reflectance separately. Is there a similar task whereby the authors could show a practical use of the decomposition and use it for comparison with other methods?
>
> In Tab. 2, we present the comparison of lighting reconstruction using different methods, and in the supplemental material video, we showcase the results of relighting. In the revised appendix, we demonstrate the relighting comparison results for other synthetic scenes in Appx Fig 20.

---

### Official Review · Reviewer_c69R · 2023-11-02

**Soundness:** 3 good
**Presentation:** 3 good
**Contribution:** 3 good
**Rating:** 6
**Confidence:** 4

**Summary:**

This paper deals with the problem of jointly reconstructing surface geometry and material properties and inferring illumination from posed images captured from multiple viewpoints in a stationary scene. The proposed approach for this task is able to deal with glossy and specular objects quite well. The underlying approach is based on inverse rendering which has been studied and has shown quite a lot of promise recently. Motivated by NeuS/Geo-NeuS, the authors have proposed an implicit 3D reconstruction approach that works in three-stages. The first stage focus on accurate surface geometry recovery even when non-Lambertian surfaces are present and the subsequent stages involve estimating the lighting directions and visibility and BRDFs. The paper presents novel ideas and presents ablation experiments to justify the importance of the proposed ideas. The overall approach outperforms existing approaches on some hand selected scenes from the DTU and SK3D benchmark.

**Strengths:**

The paper presents strong results that improve upon recent works addressing the joint geometric reconstruction and appearance modeling task – PhySG (Zhang et al. 2021), NVDiffRec and IndiSG, on appropriate datasets. Standard metrics are used for evaluating the geometric accuracy and the re-rendered images based on the reconstructed geometry and appearance models are evaluated based on PSNR.

The formulation is principled and in particular the use of surface rendering as well as volumetric rendering to deal with non-Lambertian surfaces is novel, I think. Also, training a neural network to predict the specula albedo map on top of the BRDF network is also a technical contribution to the best of my knowledge.
The SDF-based reconstructed geometry is robust and works well on non-Lambertian surfaces where Geo-NeuS and NeuS both produce noticeable artifacts.

The ablation study reported confirms that the specular albedo estimation improves accuracy in most cases on various datasets and also the second experiment confirms the importance of combining volumetric and surface rendering.

**Weaknesses:**

Although the paper reports quantitative results (Chamfer-distance based) on the DTU and SK3D datasets, they only do so for 4 and 5 scenes from DTU and SK3D respectively. It would have been more convincing to see the quantitative results on the whole dataset. The approach is sound and should technically work on all scenes and not just on scenes with glossy objects. Therefore, including these results would be informative to confirm that the more complex image formation model does not lower performance on scenes with Lambertian objects.

The approach is quite complex and involves training several different neural networks and setting hyperparameters in different stages. While Section 3.1 is well written and provides more details, I found Sections 3.2 and 3.3 to be quite short and difficult to follow. The technical presentation is sound, but the work will be difficult to reproduce due to lack of sufficient detail, especially in Section 3.3. It will help the reader if the author present pseudo code for the full approach and clarify which modules are trained and how the associated parameters are initialized and how the various hyperparameters need to be set or tuned.

The images in the paper are very small (Figures 3, 4 and 5) making it difficult to compare the results from various methods.

**Questions:**

I would appreciate if the authors could clarify the technical novelty especially around the specular albedo estimation, and how this part of their approach differs from the work of Zhang et al 2022b (IndiSG).

There are several hyperparameters in the three stages. How are they all determined? Which ones need to be tuned from scene to scene, or when dealing with different datasets?

Why is the approach evaluated only on masked objects?

I would appreciate it if the authors computed the metrics for the whole DTU dataset and showed the accuracy gap with NeuS and Geo-NeuS on all the scenes.

---

> ### Author Response · Authors · 2023-11-23
> **Authors response**
>
> We sincerely appreciate the thoughtful comments provided by the reviewers and we respond to each question in the following.
>
>
> >[W1] It would have been more convincing to see the quantitative results on the whole dataset. Including these results would be informative to confirm that the more complex image formation model does not lower performance on scenes with Lambertian objects.
>
> In the appendix, we validated our method on the whole DTU dataset in Appx Tab 7 and observed improvements in the reconstruction Chamfer distance compared to NeuS.
>
> >[W2] The approach is quite complex and involves training several different neural networks and setting hyperparameters in different stages. How are they all determined? Which ones need to be tuned from scene to scene, or when dealing with different datasets? While Section 3.1 is well written and provides more details, I found Sections 3.2 and 3.3 to be quite short and difficult to follow.
>
> In the original submission, the hyperparameters and details  of our 2nd and 3rd stages are set in the appendix section B and D respectively in the supplementary material. We have revised and incorporated the appendix at the end of the references now and further added section E for stage 3.
>
> >[W3] The images in the paper are very small (Figures 3, 4 and 5) making it difficult to compare the results from various methods.
>
> Thank you for your suggestion. We have placed the large figures on an anonymous website for viewing and comparison at https://authors-hub.github.io/Factored-NeuS.
>
> >[Q1] I would appreciate if the authors could clarify the technical novelty especially around the specular albedo estimation, and how this part of their approach differs from the work of Zhang et al 2022b (IndiSG).
>
> In the first stage, we proposed a novel surface reconstruction method. The volume rendering module is better suited to radiance reconstruction and the surface rendering module ensures higher quality specular and diffuse separation close to the surface. Both working together enable accurate and efficient reconstruction of glossy object surfaces. Previous methods consider only using volume rendering (NeuS, VolSDF) or surface rendering IDR(PhySG, IndiSG) to reconstruct geometric information, and cannot handle the reconstruction of glossy objects, as shown in Tab. 1, Fig. 3, and Appx Fig. 7. In the second and third stages, we present continuous light visibility supervision based on SDF and a specular albedo estimation method based on spherical Gaussians. Although these components are simple, the great effectiveness of our way of utilizing them is evident in Tab. 2, Fig. 4, and Appx Fig. 10.
> Therefore, our work contains new components and also builds on existing ideas. In addition, originality also comes from how the ideas are combined and what problem it solves. In our case, we built a method to do inverse rendering for glossy objects in an unsupervised manner. It is something that previous methods fail on this task.
>
> >[Q2] Why is the approach evaluated only on masked objects?
>
> We did not use masks when conducting experiments on real dataset DTU and SK3D. Methods like IndiSG use masks when running synthetic datasets. To maintain consistency, we also use masks when running synthetic datasets.
>
> >[Q3] I would appreciate it if the authors computed the metrics for the whole DTU dataset and showed the accuracy gap with NeuS and Geo-NeuS on all the scenes.
>
> Regarding Geo-NeuS, the method utilizes additional point cloud supervision  to improve surface reconstruction results, and we were unable to reproduce the original paper's average result of 0.508 using their code as the github issue, as mentioned in the GitHub issue. We have compared our method with NeuS on the entire DTU dataset in Appx Tab 7 and found that our approach exhibits overall improvement in performance as shown in the appendix.

---

### Official Review · Reviewer_4mZH · 2023-11-05

**Soundness:** 2 fair
**Presentation:** 2 fair
**Contribution:** 2 fair
**Rating:** 3
**Confidence:** 4

**Summary:**

This paper proposes a three-stage method to reconstruct geometry, material and lighting of glossy objects from multi-view posed images. They reconstruct the scene radiance and signed distance function (SDF) in the first stage with both volume and surface rendering, and decompose the color into diffuse and specular components for surface rendering. They then distill and model light visibility and indirect illumination from the learned SDF and radiance field. Finally, they perform material and direct illumination estimation based on the learned geometry and visibility/indirect light. Experimental results show that they can recover more plausible surface geometry and albedo of glossy objects compared to existing methods.

**Strengths:**

- This paper proposes to progressively decompose the glossy object into plausible geometry, material and illumination. They model the visibility and indirect light according to the recovered geometry and radiance field, and perform the material reconstruction based on the geometry and indirect light. The progressive design facilitates the decomposition of the albedo/specular and the illumination.
- Experimental results show the proposed method successfully recovers the surface of the glossy objects and decomposes the diffuse/specular components.

**Weaknesses:**

- Lack of reference to NeRO [a], DIP [b]. NeRO [a] proposes a two-step approach for reconstructing the geometry and the BRDF of reflective objects with strong reflective appearances. They also introduce real/synthetic glossy dataset for evaluation. DIP [b] proposes a physics-based interreflection-aware illumination model and end-to-end learns the illumination, geometry and materials. Both are released earlier and with code available. The authors should compare the proposed method with them.
- Lack of novelty. The proposed method consists of three stages which combine ideas from different papers. For the first stage, they propose to perform joint volume and surface rendering, which is similar to S^3-NeRF [c], and joint appearance and BRDF modeling which follows TensoIR [d], decompose color into explicit diffuse and specular components for glossy surfaces which follows Ref-NeRF[e]. The visibility and indirect light modeling in second stage mostly follow NeRFactor [f] and IndiSG [g]. The BRDF modeling in last stage follows PhySG [h].
- Evaluations/analysis are not thorough. For instance, the diffuse/specular component of the first stage is not visualized/analyzed. There is also no ablation study on the design of the modeling of the two components. Besides, it would be better to visualize the geometry/albedo/specular/roughness/visibility/indirect light/environment map at the same time for all the datasets. Most results in the paper are partially shown for different datasets. Visibility and indirect lights are only included in the videos of three objects without any analysis. Environment maps are only visualized in one ablation study in supplementary.
- The progressive decomposition is not end-to-end, the geometry is fixed after first stage.
- Symbols used are a bit messy.

[a] Liu, Yuan, et al. "NeRO: Neural Geometry and BRDF Reconstruction of Reflective Objects from Multiview Images." SIGGRAPH 2023
[b] Deng, Youming, et al. "DIP: Differentiable Interreflection-aware Physics-based Inverse Rendering." 3DV 2024
[c] Yang, Wenqi, et al. "S^3-NeRF: Neural Reflectance Field from Shading and Shadow under a Single Viewpoint." NeurIPS 2022
[d] Jin, Haian, et al. "TensoIR: Tensorial Inverse Rendering." CVPR 2023
[e] Verbin, Dor, et al. "Ref-nerf: Structured view-dependent appearance for neural radiance fields." CVPR 2022
[f] Zhang, Xiuming, et al. "Nerfactor: Neural factorization of shape and reflectance under an unknown illumination." ACM ToG 2021
[g] Zhang, Yuanqing, et al. "Modeling indirect illumination for inverse rendering." CVPR 2022
[h] Zhang, Kai, et al. "Physg: Inverse rendering with spherical gaussians for physics-based material editing and relighting." CVPR 2021

**Questions:**

- The datasets for evaluation should be more diverse. The lighting of SK3D dataset is not complex and the backgrounds are all black. The proposed method should also be evaluated on the Glossy-Blender dataset and Glossy-Real dataset proposed by NeRO [a] whose surfaces are more reflective.
- What’s the performance on objects that are diffuse or not very glossy?
- Better visualize the pipeline and models of all stages for easier understanding.
- The proposed method chooses simplified modeling for diffuse and specular component in the first stage, where diffuse does not consider shading and specular and only consider reflection direction and normal as input of MLP. Since the geometry will be fixed in the following two stages, how would the design affect the geometry reconstruction?

---

> ### Author Response · Authors · 2023-11-23
> **Authors response (part 1/3)**
>
> We sincerely appreciate the thoughtful comments provided by the reviewers and we respond to each question in the following.
>
> > [W1] Lack of reference to NeRO [a], DIP [b]. The authors should compare the proposed method with them.
>
> We appreciate the reviewer's feedback regarding the lack of reference to NeRO [a] and DIP [b]. We duly incorporate references to NeRO [a] and DIP [b] in our revised manuscript. For comparison, we compared our method with the already published work NeRO. Our method is compared with NeRO on the established real datasets DTU and SK3D for evaluating glossy objects. Our findings demonstrate that NeRO performs less effectively than our approach on real glossy datasets DTU in Appx Fig. 17. NeRO not only struggles to accurately restore detailed information but also fails to address the negative impact of partial highlights on the geometry. Moreover, the presence of shadows causes NeRO to mistakenly reconstruct shadowed areas as real objects and fill them in (bricks and skull models). In addition, our quantitative evaluation of Chamfer distance for the DTU dataset performs better as presented in Appx Tab. 7. When running the SK3D dataset, similar issues were encountered as in the DTU. Even in scenarios with a simple background and straightforward geometry, NeRO still tends to lose certain details and fills in shadowed areas (flower pot model) as shown in Appx Fig. 18. In addition, our quantitative evaluation of Chamfer distance for the SK3D dataset performs better as presented in Appx Tab. 8. Furthermore, we extended our comparison to include new glossy datasets, Glossy-Blender dataset and Glossy-Real dataset in Appx Fig 19 and Appx Tab 9, where although NeRO performs better, but our method is also capable of mitigating the impact of highlights on geometry. Our method demonstrated comparable results to NeRO. The experimental results have been presented in the Appendix. DIP is a concurrent work that will be published in 3DV 2024. We added a citation, but we think the paper only establishes that DIP looks good on synthetic data and does not claim that the method can handle glossy objects.
>
> > [W2] Lack of novelty.
> > [W2 / 1] For the first stage, they propose to perform joint volume and surface rendering, which is similar to S^3-NeRF [c]
>
> We respectfully disagree with this observation. To address the degradation issue in the model training caused by the reduced sampling range during the transition from volume rendering to surface rendering in UNISURF, S^3 NeRF defines volume rendering as the integration of surface rendering colors on the ray to increase the sampled region. This approach is fundamentally different in implementation and motivation from ours. When we applied the construction method of S^3 NeRF, which involves integrating surface rendering colors into volume rendering, to modify our model structure, we found that this modeling approach did not eliminate the impact of highlights on the surface reconstruction results. The experimental quantitative and qualitative results have been shown in Appx Fig 11 in the appendix. In addition, we exhibit differences in many other aspects, such as our use of Signed Distance Function (SDF) instead of an occupancy field to represent the implicit function and determine the position of surface points.
>
> > [W2 / 2] For the first stage, joint appearance and BRDF modeling which follows TensoIR [d]
>
> At the first stage, the decomposition of diffuse and specular is not a true BRDF model. This is because the MLP in the first stage is used solely for predicting the components of diffuse and specular reflection, rather than predicting material properties such as albedo and roughness. The decision to directly predict colors instead of material properties in the first stage serves two purposes: reducing model complexity by focusing on the direct prediction of specular reflection color, and optimizing geometry for better reconstruction. By decomposing highlights through the network in the first stage, surfaces with specular reflections can be reconstructed more effectively, demonstrated by the presence of flower pot ablation in Fig 5, and without encountering the concavity issues observed in other methods. No prior work has utilized this method for geometry (SDF) optimization. After the completion of the first stage, where we believe the model's geometry has been optimized, the third stage focuses on predicting materials (BRDF model) based on the optimized geometry. During this process, the geometry model remains fixed and unchanged.

---

> ### Author Response · Authors · 2023-11-23
> **Authors response (part 2/3)**
>
> > [W2 / 3] For the first stage, decompose color into explicit diffuse and specular components for glossy surfaces which follows Ref-NeRF[e].
>
> A distinction between our method and Ref-NeRF is that we use surface point colors for both diffuse and specular components, whereas Ref-NeRF utilizes volume rendering colors for diffuse and specular components. In our ablation study, we demonstrate that using volume rendering colors does not completely eliminate the impact of highlights on the geometric reconstruction of the flower pot. Additionally, Ref-NeRF relies on a density field, while our approach is based on an SDF field, allowing us to obtain smoother 3D shape models. (When using an SDF field, the incorporation of additional feature vectors from the SDF network output into the color prediction model is necessary.)
>
> > [W2 / 4] The visibility and indirect light modeling in the second stage mostly follow NeRFactor [f] and IndiSG [g].
>
> We have indeed incorporated the concept of continuous light visibility from NeRFactor to enhance the binary light visibility of IndiSG, which predicts light visibility as either 0 or 1. We predict light visibility as 1-Σw. Existing ablation studies in Tab 3 and Appx Fig 10 demonstrate that employing this approach yields better results. Indirect lighting is currently a widely used method for lighting modeling, and we do not claim novelty in this aspect. However, there are many possible ways to construct a complete system and it is not trivial to arrange all parts into a well working system, even if some of these parts have been used before and are mainly adapted.
>
> > [W2 / 5] The BRDF modeling in last stage follows PhySG [h].
>
> In the third stage, our contribution is the introduction of specular albedo, a feature absent in PhySG. Through analysis, we discovered that incorporating specular albedo enhances the reconstruction of material properties and rendering colors.
>
> > [W3] Evaluations/analysis are not thorough.
> > [W3 / 1]  the diffuse/specular component of the first stage is not visualized/analyzed. There is also no ablation study on the design of the modeling of the two components.
>
> We actually think our experiments are quite extensive, especially compared to other papers in the field. In the original submission, we presented an ablation study in Fig. 5 to demonstrate the impact of designing the two components on shape reconstruction for the first stage.
> Additionally, our goal in the first stage is the reconstruction of well-defined geometric shapes for glossy objects rather than generating diffuse/specular components. Nevertheless, we showcase the decomposed results of diffuse and specular components in Appx Fig 12 in the appendix, demonstrating a reasonable outcome of the decomposition process.
>
> > [W3 / 2] It would be better to visualize the geometry / albedo / specular / roughness / visibility / indirect light / environment map at the same time for all the datasets. Environment maps are only visualized in one ablation study in supplementary.
>
> Thank you for your suggestion. We have added these components for all indiSG, DTU and SK3D datasets in Appx Fig 13, Fig 14 and Fig 15 in the appendix.
>
> > [W3 / 3] Visibility and indirect lights are only included in the videos of three objects without any analysis.
>
> In the original submission, we have included the ablation study for visibility in Appx Fig 10 and analyse the necessity of including the visibility components. In the appendix, we additionally showcase ablation experiments on indirect light in Appx Fig 16. Through analysis, it is evident that the reconstruction quality of materials and rendering deteriorates in the absence of indirect light.
>
> > [W4] The progressive decomposition is not end-to-end, the geometry is fixed after the first stage.
>
> In the first stage, the decomposition of diffuse and specular components contributes to the improvement in geometric quality, as evidenced by flower pot ablation. After the training in the first stage, if the model parameters are not fixed and the subsequent 2nd and 3rd stages are executed, the geometric quality experiences a decline due to the increased complexity of model training. Not being end-to-end is a disadvantage in some sense, but we obtain better quality with our current method which is an advantage.

---

> ### Author Response · Authors · 2023-11-23
> **Authors response (part 3/3)**
>
> > [W5] Symbols used are a bit messy.
>
> We appreciate the reviewer's feedback on the symbols used in our manuscript. We respectfully disagree with the notion that our symbols are messy.
> We use the IndiSG paper as the starting point for our notation, but we also introduced several changes. We had been making multiple revisions to our symbol usage while drafting the manuscript, with a dedicated focus on improving clarity and coherence. We believe that the current set of symbols accurately represents the technical aspects of our work and facilitates a comprehensive understanding of the methodologies employed.
> At the same time, we are fully open to further enhancing our manuscript. If the reviewer could provide specific instances or aspects of our symbol usage that they find unclear or confusing, we would be more than willing to address these concerns in a targeted manner. Such specific feedback would be invaluable in helping us refine our manuscript to better meet the needs of our audience. We are committed to making our research as clear and understandable as possible and welcome any constructive feedback to achieve this goal.
>
> > [Q1] The datasets for evaluation should be more diverse. The lighting of SK3D dataset is not complex and the backgrounds are all black. The proposed method should also be evaluated on the Glossy-Blender dataset and Glossy-Real dataset proposed by NeRO [a] whose surfaces are more reflective.
>
> We presented results on the Glossy-Blender dataset and Glossy-Real dataset in Appx Fig 19 and Appx Tab 9, demonstrating comparable outcomes to NeRO. It is worth clarifying that our method performs better than NeRO on the real-world DTU and SK3D datasets.
>
> > [Q2] What’s the performance on objects that are diffuse or not very glossy?
>
> In the original submission, we presented material decomposition results for diffuse objects such as hotdog in Fig 4 and Tab 2. In the revised paper, we further showcase results for the entire DTU dataset in Appx Tab 7 and Fig 17, which includes many objects that are diffuse or not very glossy. Our method is also well-suited to handle this type of data.
>
> > [Q3] Better visualize the pipeline and models of all stages for easier understanding.
>
> Thank you for your suggestion. In the revised version, we have illustrated all stages of the model in Fig.1 and Appx Fig.6.
>
> > [Q4] The proposed method chooses simplified modeling for diffuse and specular component in the first stage, where diffuse does not consider shading and specular only consider reflection direction and normal as input of MLP. Since the geometry will be fixed in the following two stages, how would the design affect the geometry reconstruction?
>
> This is one of the main insights from our work. If the lighting and material model is too complex, the surface reconstruction will be unstable and overfit. If the lighting and material model is too simple, the surface reconstruction will interpret glossy reflections as bumps and grooves as shown in Fig.5 in the paper. We design a surface reconstruction method that is robust in real scenes which is a trade-off between being too complex and overly simplistic. We experimented with many different versions of simpler and more complex material and lighting models for surface reconstruction and this is the best one we found.

---

### Meta-Review · Area_Chair_UxuA · 2023-12-06

**Metareview:**

This paper presents a method for jointly reconstructing surface geometry, material properties, and illumination from posed images captured from multiple viewpoints in a static scene. The method consists of three stages to progressively decompose the shape, material, and illumination. The experiments show promising reconstruction results.

The major strengths of the paper are:
(1) The three-stage progressive decomposition approach is somehow new.
(2) Experimental results show successful reconstructions, particularly effective for glossy objects.

On the other hand, the chief weakness pointed out by reviewers is that the novelty of the paper is rather limited due to that the paper combines multiple existing ideas/works.

The reviewers and AC see the merit of this paper. However, there was a remaining concern about the novelty of the work, i.e., the paper combines multiple existing ideas/works. The authors' rebuttal addressed the issue to some extent; however, it did not fully resolve it. The reviewers and AC have discussed after reading the rebuttal and reached the final recommendation.

**Justification For Why Not Higher Score:**

There still remained a concern about the novelty of the work. The reviewers and AC agree that the authors have incorporated some modifications during the adoption. However, it was still regarded that the novelty of the paper was limited.

**Justification For Why Not Lower Score:**

N/A

---

### Decision · Program_Chairs · 2024-01-16

Reject